# Rescuing SERCA2 pump deficiency improves bone mechano-responsiveness in type 2 diabetes by shaping osteocyte calcium dynamics

Xi Shao [1,7], Yulan Tian[1,7], Juan Liu[1,7], Zedong Yan[1,7], Yuanjun Ding [1], Xiaoxia Hao[1], Dan Wang[1], Liangliang Shen[2], Erping Luo[1], X. Edward Guo[3], Peng Luo [4] ✉, Wenjing Luo[5] ✉, Jing Cai[6] ✉ & Da Jing [1,5] ✉

Type 2 diabetes (T2D)-related fragility fractures represent an increasingly tough medical challenge, and the current treatment options are limited. Mechanical loading is essential for maintaining bone integrity, although bone mechano-responsiveness in T2D remains poorly characterized. Herein, we report that exogenous cyclic loading-induced improvements in bone architecture and strength are compromised in both genetically spontaneous and experimentally-induced T2D mice. T2D-induced reduction in bone mechano-responsiveness is directly associated with the weakened $Ca^{2+}$ oscillatory dynamics of osteocytes, although not those of osteoblasts, which is dependent on PPARα-mediated specific reduction in osteocytic SERCA2 pump expression. Treatment with the SERCA2 agonist istaroxime was demonstrated to improve T2D bone mechano-responsiveness by rescuing osteocyte $Ca^{2+}$ dynamics and the associated regulation of osteoblasts and osteoclasts. Moreover, T2D-induced deterioration of bone mechano-responsiveness is blunted in mice with osteocytic SERCA2 overexpression. Collectively, our study provides mechanistic insights into T2D-mediated deterioration of bone mechano-responsiveness and identifies a promising countermeasure against T2D-associated fragility fractures.

Type 2 diabetes (T2D), which accounts for the vast majority of all diabetes cases and afflicts over 400 million people worldwide[1], is characterized by chronic metabolic abnormalities with persistent hyperglycemia due to insulin resistance, thereby detrimentally affecting multiple organs, including the heart, kidneys, and eyes.

Furthermore, there is growing recognition and evidence of T2D-induced adverse effects on the skeletal system[2]. In contrast to non-diabetics, significant increases in cortical porosity and declines in trabecular and cortical material properties have been observed in patients with T2D, which contribute to heightening the risk of

[1]Department of Biomedical Engineering, Fourth Military Medical University, Xi'an, China. [2]The State Key Laboratory of Cancer Biology, Department of Biochemistry and Molecular Biology, Fourth Military Medical University, Xi'an, China. [3]Bone Bioengineering Laboratory, Department of Biomedical Engineering, Columbia University, New York, NY, USA. [4]Department of Neurosurgery, Xijing Hospital, Fourth Military Medical University, Xi'an, China. [5]The Ministry of Education Key Lab of Hazard Assessment and Control in Special Operational Environment, Fourth Military Medical University, Xi'an, China. [6]College of Basic Medicine, Shaanxi University of Chinese Medicine, Xianyang, China. [7]These authors contributed equally: Xi Shao, Yulan Tian, Juan Liu, Zedong Yan. ✉e-mail: pengluo@fmmu.edu.cn; luowenj@fmmu.edu.cn; 1988cai@163.com; jingdaasq@126.com

subsequent osteoporotic fractures[3–6]. Numerous clinical studies have reported that T2D is associated with a significant enhanced risk of bone fractures, particularly in bones of the weight-bearing lower extremities (e.g., tibia, femur, and hip)[7,8]. Importantly, subsequent to the occurrence of a fracture, a series of post-injury issues, including poor wound and fracture healing, a heightened risk of infection, and delayed recovery, can potentially lead to significant increases in morbidity, mortality, and related healthcare costs[9]. However, to date, the potential therapeutic interventions against T2D-related fragility fractures have rarely been reported.

Mechanical loading is essential for the maintenance of bone homeostasis and skeletal integrity, and abundant evidence has accumulated to indicate that the mechanical stimulation of skeletons (e.g., resistance training and aerobic exercise) induces a pronounced increase in bone mass and strength by promoting bone formation and suppressing bone resorption[10,11]. In contrast, a lack of weight-bearing forces, such as that associated with long-term therapeutic bed rest and spaceflight, can contribute to rapid bone loss[12,13]. These findings accordingly highlight the importance of mechanical adaptation in the control of bone health, and indicate that augmenting bone mechano-responsiveness may become a key strategy in counteracting osteoporosis[14,15]. However, to the best of our knowledge, the characteristics of the response and transduction of the skeletons of patients with T2D to external mechanical stimuli have not been studied previously.

Osteocytes, comprising the overwhelming majority of all bone cells (~95%), are encapsulated within a fluid-filled mineralized bone matrix (the lacunar-canalicular system, LCS), which forms an extensive neuron-like interconnected network[16]. Through the fluid transport in the LCS, osteocytes absorb nutrients and oxygen, and communicate with other osteocytes, surface cells, and marrow cells[17]. Although initially considered to be inert placeholders, growing evidence obtained in recent years indicates that osteocytes may function as dominant mechanosensory cells in bone[18], and also play roles as master orchestrators of bone homeostasis by controlling the responses of osteoclasts and osteoblasts to both mechanical and hormonal cues[19]. Indeed, several recent studies have shown the high sensitivity of osteocytes to a high-glucose/high-fat (HGHF) environment, increased apoptosis, altered cytokine expression (e.g., sclerostin and RANKL), and impaired cell morphology and network connectivity[20,21]. Thus, targeting different facets of osteocyte biology represents an emerging therapeutic approach for preventing various types of bone fragility, including that associated with T2D[22].

A major obstacle to the study of osteocytes is our current lack of understanding regarding the mechanisms whereby osteocytes respond to and transduce mechanical strain. In the body, calcium ($Ca^{2+}$) functions as an essential and ubiquitous second messenger that contributes to the regulation of diverse cellular activities. In particular, in contrast to constant $Ca^{2+}$ signaling, the oscillatory changes in cytosolic $Ca^{2+}$ are considered to be advantageous with respect to enhancing the efficiency and specificity of gene expression, thereby regulating numerous subsequent cellular processes[23,24]. In our previous studies, we found that skeletons detect and respond to physiological mechanical loading via LCS fluid shear stress-induced osteocyte $Ca^{2+}$ oscillations, characterized by unique repetitive robust $Ca^{2+}$ spikes[25,26]. Osteocytes were also found to be considerably more responsive in $Ca^{2+}$ signaling than osteoblasts under mechanical stimulation[25,26]. Moreover, the mechanically induced $Ca^{2+}$ oscillations were found to induce immediate actin network contractions, thereby driving the secretory activity of extracellular vesicles containing bone regulatory proteins (e.g., RANKL, OPG, and sclerostin)[27]. Thus, it is reasonable to speculate that the manipulation of osteocyte $Ca^{2+}$ signatures may provide a unique approach for improving bone mechano-responsiveness.

In this study, we hypothesize that bone mechano-responsiveness is compromised by T2D, which is attributed to the altered $Ca^{2+}$ oscillatory dynamics in osteocytes. Based on a unilaterally constrained hindlimb loading model, we show that load-induced increases in bone quantity and quality are abolished in both genetically spontaneous (KK-Ay mice) and experimentally-induced (high-fat diet/streptozotocin-treated mice, HFD + STZ) T2D models. Using unique multi-scale $Ca^{2+}$ probing techniques, we observed that T2D-associated reductions in bone mechano-responsiveness are primarily attributed to an attenuation of $Ca^{2+}$ dynamics in osteocytes, although not in osteoblasts. Mechanistically, the peroxisome proliferator-activated receptor α (PPARα)-mediated specific reduction in the sarco/endoplasmic reticulum $Ca^{2+}$-ATPase 2 (SERCA2) pump contributes to the attenuated osteocyte $Ca^{2+}$ oscillatory dynamics in T2D skeletons, characterized by fewer and weaker $Ca^{2+}$ spikes with prolonged $Ca^{2+}$ uptake. Treatment with the SERCA2 agonist istaroxime (ISTA) was found to improve bone mechano-responsiveness in both KK-Ay and HFD + STZ mice by rescuing osteocyte $Ca^{2+}$ dynamics and the associated $Ca^{2+}$-mediated regulation of osteoblasts and osteoclasts. Moreover, we detected a suppression of the T2D-induced deterioration of bone mechano-responsiveness in mice with overexpression of osteocytic SERCA2.

## Results

### Exogenous cyclic loading-induced increases in bone mass and strength are suppressed in both genetically spontaneous and experimentally-induced T2D mice

We initially investigated whether exogenous cyclic loading-induced anabolic changes in bone quantity and quality were impaired in T2D mice. KK-Ay mice were used as a genetically spontaneous T2D model, which was generated by transferring the yellow obese gene (Ay allele) into diabetic KK mice[28]. Additionally, an experimentally-induced T2D mouse model was established by injecting mice with low-dose STZ for five consecutive days combined with HFD feeding[29]. Both groups of model mice exhibited typical T2D symptoms, characterized by obesity, elevated food intake and urine output, and hyperglycemia (Fig. S1). Having successfully established the T2D models, the left tibiae of these male mice in the non-diabetes, KK-Ay, and HFD + STZ groups were subjected to axial cyclic compressive loading for 2 weeks, with peak loads of 9.0, 6.8, and 7.9 N (corresponding to 1455 με tensile strain on tibial antemedial surface), respectively (Fig. 1a–c, Fig. S2). As within-group controls, the right tibiae of mice were not subjected to cyclic compressive loading. According to the results of micro-CT scanning and H&E staining, the mechanically loaded tibiae in male non-diabetic mice exhibited a significant increase (approximately 30%) in trabecular bone mass and thickness compared with the unstimulated contralateral tibiae (Fig. 1d, e). In contrast, we found that exogenous cyclic loading resulted in no significant improvements in tibial trabecular bone mass or microarchitecture in either male KK-Ay or HFD + STZ mice (Fig. 1d, e). Exogenous cyclic loading on tibiae significantly reduced the number of bone marrow adipocytes in male non-diabetic mice, but not in either KK-Ay or HFD + STZ mice (Fig. 1e and Fig. S1). Moreover, in non-diabetic mice, we observed that exogenous cyclic loading promoted a significant increase in cortical bone thickness and suppressed cortical porosity (Fig. 1f, g). In contrast, in both KK-Ay and HFD + STZ mice, we detected no significant changes in cortical bone thickness or porosity between the loaded and control tibiae (Fig. 1f, g). Furthermore, the results of biomechanical three-point bending and nanoindentation testing revealed that exogenous cyclic loading-induced increases in whole-bone mechanical properties and the intrinsic material properties of tibiae were significantly suppressed in KK-Ay mice and HFD + STZ mice (Fig. 1h, i). The results of behavioral open field tests revealed no significant difference in horizontal or vertical activity between the non-diabetic mice and the two T2D mouse models (Fig. S3).

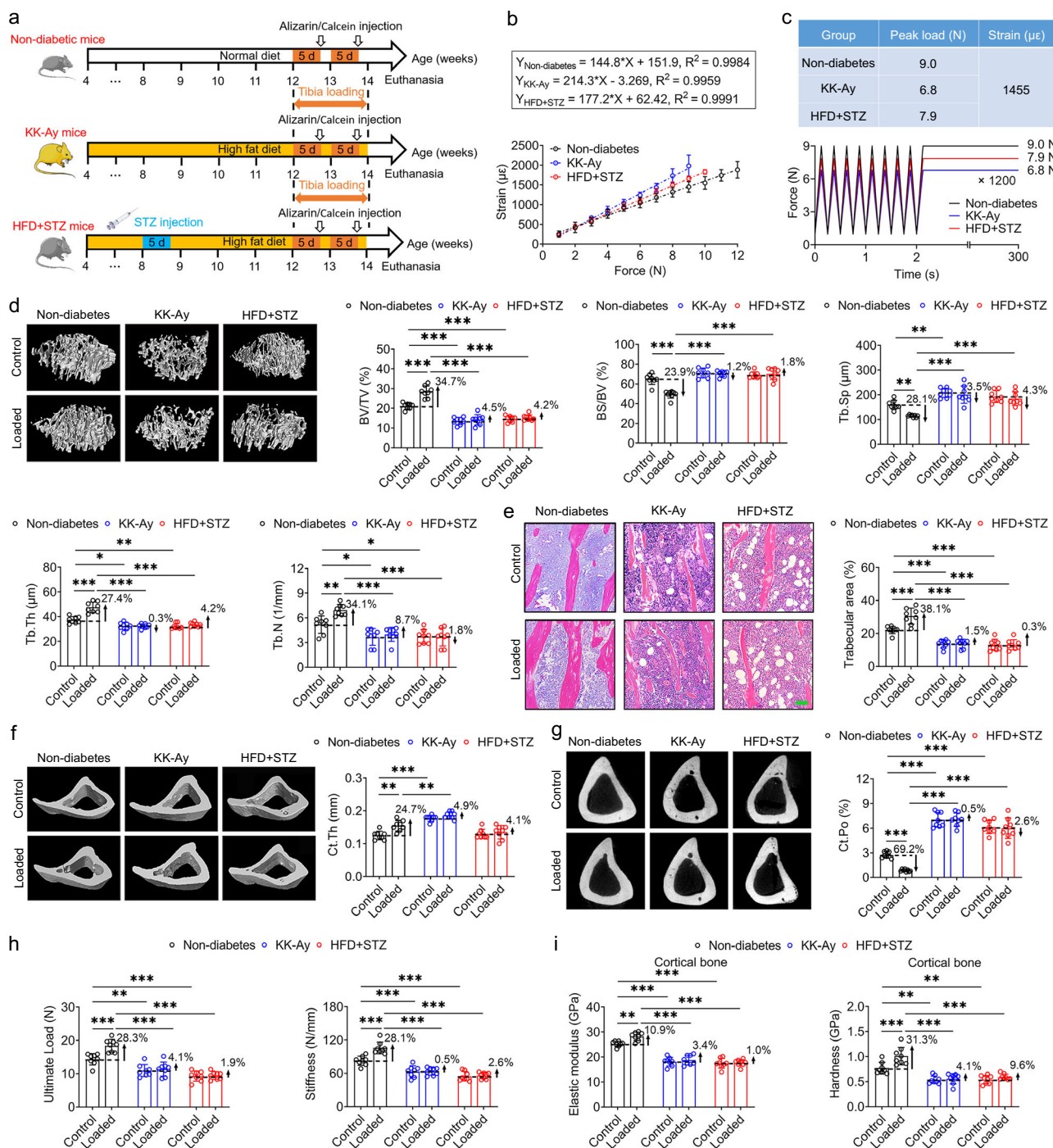

**Fig. 1 | Exogenous cyclic loading-induced increases in bone mass and strength are suppressed in both genetically spontaneous and experimentally-induced T2D mice. a** The flow diagram of the experimental protocol, including the establishment of the genetically spontaneous and experimentally-induced male T2D mice, i.e., the KK-Ay mice and high-fat diet/streptozotocin-treated (HFD + STZ) mice, and subsequent application of cyclic compressive mechanical loading on tibiae for 2 weeks. The left tibia was subjected to daily physiological mechanical loading with 1200 cycles/day, and the contralateral right tibia was not subjected to cyclic loading and served as a control. **b** The linear correlation between the applied loads and strains on the antemedial surface of the mouse tibia in the non-diabetes, KK-Ay, and HFD + STZ groups. **c** A waveform of cyclic ramp loading at 4 Hz applied with the identical tensile strain (1455 με) generated on the antemedial

surface by applying cyclic compressive loading with 9.0 N, 6.8 N, and 7.9 N on the mouse tibia in the non-diabetes, KK-Ay, and HFD + STZ groups, respectively. **d** Three-dimensional reconstructed micro-CT results showing proximal tibial trabecular bone microarchitecture in mice of the non-diabetes, KK-Ay and HFD + STZ groups. **e** Two-dimensional hematoxylin-eosin (H&E) staining images showing trabecular area microstructure in the proximal tibia. **f, g** Micro-CT results showing cortical bone thickness and cortical porosity. **h, i** Comparison of the whole-bone mechanical properties via the three-point bending testing and local tissue-level material properties via the nanoindentation testing. Graphs represent mean ± SD ($n = 8$ mice per group). *$P < 0.05$, **$P < 0.01$, and ***$P < 0.001$ by two-way ANOVA with Bonferroni's post test. Specific $P$ values are provided in the Source Data file. Scale bar: **e** 50 μm.

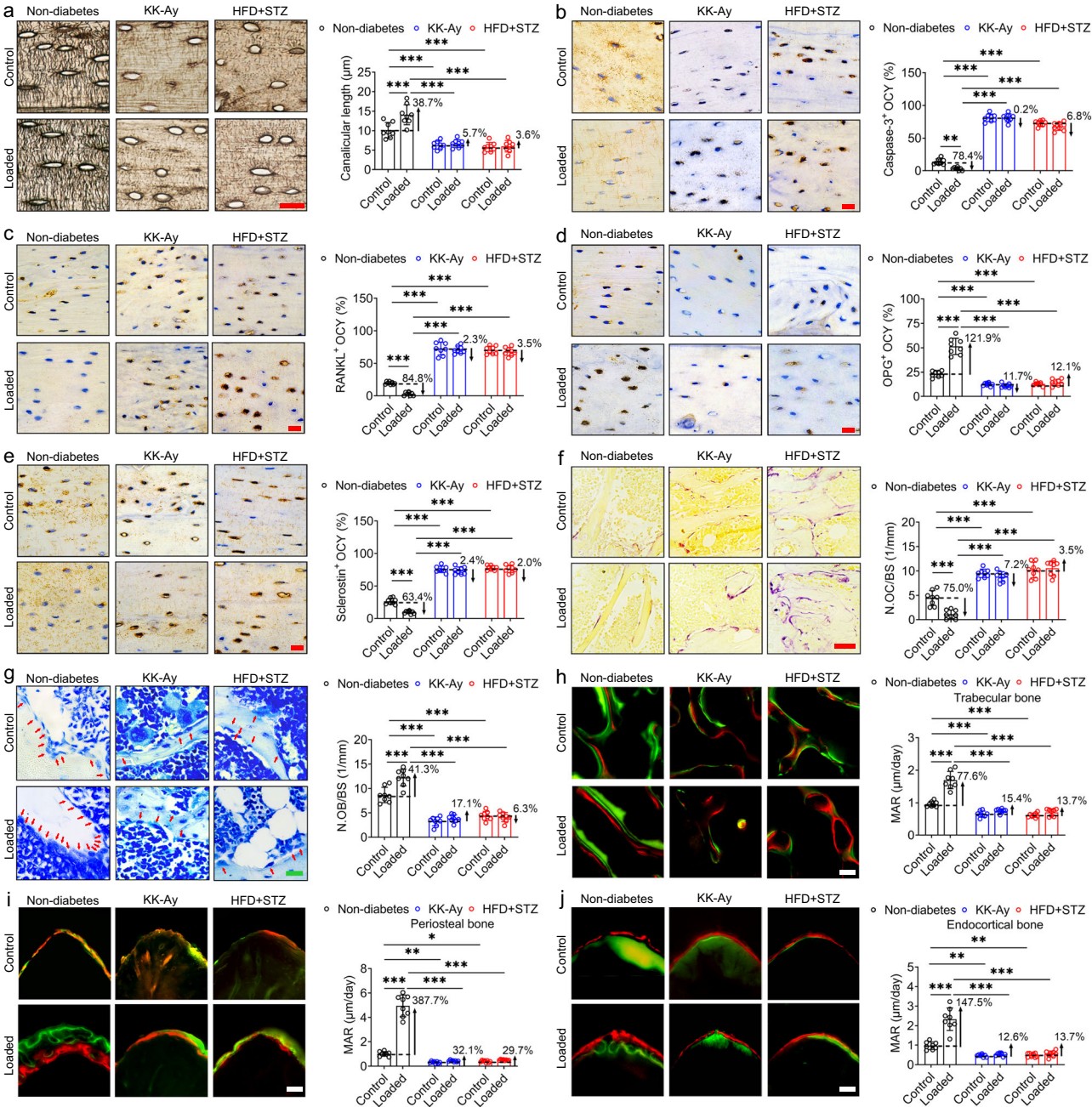

**Fig. 2 | Exogenous cyclic loading-mediated regulation of osteocytes, osteoblasts, and osteoclasts is abolished in T2D mice. a** Ploton silver staining showing the morphology of the osteocyte canalicular network in male diabetic and non-diabetic tibiae following cyclic compressive loading, and the corresponding statistical result. **b–e** Immunohistochemical staining of Caspase-3, RANKL and OPG, sclerostin expression in osteocytes (brown) in diabetic and non-diabetic tibial cortical bone. **f** TRAP staining to label osteoclasts (claret-red) in diabetic and non-diabetic tibial trabecular bone, and the corresponding statistical result of the number of osteoclast per millimeter of bone surface (N.OC/BS). **g** Toluidine blue staining to label osteoblasts on in diabetic and non-diabetic trabecular bone surface, and the corresponding statistical result of the number of osteoblast per millimeter of bone surface (N.OB/BS). **h–j** Calcein and alizarin red double labeling of the trabecular, periosteal, and endocortical bone surfaces in diabetic and non-diabetic tibiae, and the corresponding statistical results of mineral apposition rate (MAR). Graphs represent mean ± SD ($n = 8$ mice per group). *$P < 0.05$, **$P < 0.01$ and ***$P < 0.001$ by two-way ANOVA with Bonferroni's post test. Specific $P$ values are provided in the Source Data file. Scale bars: **a–e, g** 20 μm; **f, h–j** 50 μm.

## Exogenous cyclic compressive loading-mediated regulation of osteocytes, osteoblasts, and osteoclasts is abolished in T2D mice

We subsequently examined changes in the bone cells involved in bone metabolism (osteocytes, osteoblasts, and osteoclasts) in response to exogenous cyclic compressive loading in T2D mice. The findings of Ploton silver staining, immunohistochemical staining and H&E staining revealed that compared with the control tibiae of non-diabetic mice, there were significant decreases in canalicular density and length and osteocytic OPG protein expression, and significant increases in osteocytic survival rate and RANKL and sclerostin expression in the control tibiae of T2D mice (Fig. 2b–e, Fig. S1). Exogenous cyclic loading had the effect of inducing a significant increase in canalicular density and length, osteocytic survival rate and OPG protein expression, as well as a pronounced reduction in the RANKL and sclerostin protein expression in tibial osteocytes of the non-diabetic mice (Fig. 2b–e, Fig. S1). Contrastingly, we found that compared with the untreated

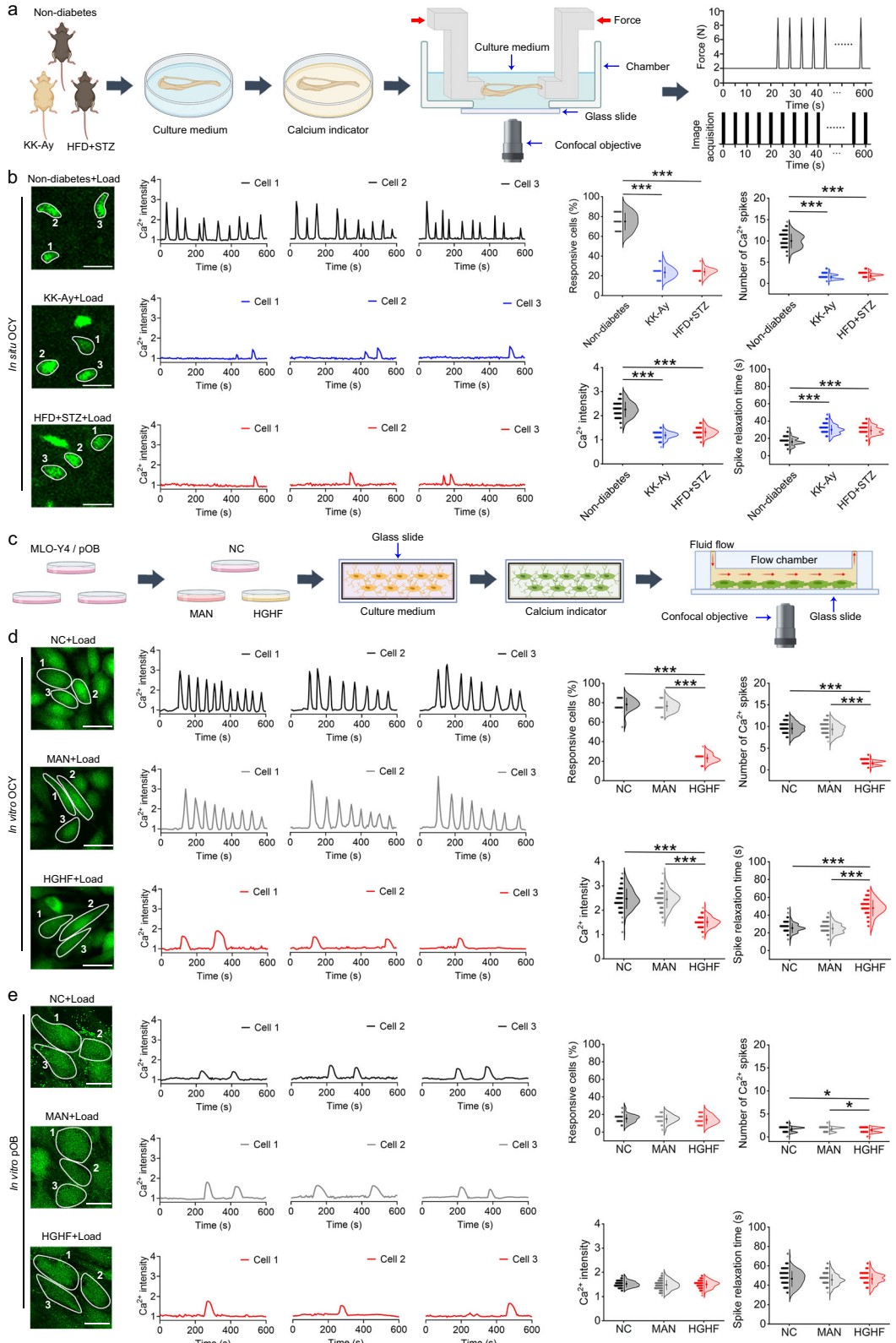

contralateral hindlimbs, there were no significant changes in canalicular density, canalicular length, or the apoptosis of osteocytes in the loaded tibiae of KK-Ay and HFD + STZ mice (Fig. 2a, b and Fig. S1). Osteocyte-associated key protein expression, including that of RANKL, OPG, and sclerostin, similarly showed no significant difference between the loaded and control tibiae in KK-Ay and HFD + STZ mice (Fig. 2c–e). Moreover, exogenous cyclic loading was found to

contribute to a significant reduction in the number of osteoclasts on tibial bone surfaces and a significant increase in the number of osteoblasts on bone surfaces and the rate of mineral apposition on trabecular, periosteal, and endocortical bone surfaces in non-diabetic mice (Fig. 2f–j). In contrast, in the two T2D animal models, compared with the control tibiae, mechanically loaded tibiae showed no significant differences with respect to the number of osteoblasts on bone

**Fig. 3 | The osteocytic Ca²⁺ oscillatory response to mechanical loading is compromised in T2D both in vitro and in situ. a** The experimental protocol of real-time osteocyte Ca²⁺ imaging in the tibiae of male T2D mice in situ subjected to cyclic compressive loading using the synchronized cyclic loading/confocal imaging technique. **b** Representative intracellular Ca²⁺ profiles of osteocytes in situ in diabetic and non-diabetic tibiae in response to cyclic mechanical loading, and the corresponding statistical results, including the percentage of responsive cells, the number of Ca²⁺ spikes, the average Ca²⁺ intensity, and the relaxation time of Ca²⁺ spikes. **c** The experimental protocol of real-time intracellular Ca²⁺ imaging in MLO-Y4 osteocytic cells and primary osteoblasts in vitro exposed to the high-glucose and high-fat (HGHF) condition for 48 h in response to subsequent fluid shear stress (FSS) stimulation. **d** Comparison of intracellular Ca²⁺ dynamics of MLO-Y4 osteocytic cells in vitro exposed to normal medium, mannitol-treated medium (MAN, osmotic pressure-matched control), and HGHF-treated medium in response to subsequent steady FSS stimulation (2 Pa). **e** Comparison of intracellular Ca²⁺ dynamics of normal, mannitol-treated, and HGHF-treated primary osteoblasts in vitro under steady FSS stimulation (2 Pa). Graphs represent mean ± SD ($n = 120$ cells per group). *$P < 0.05$ and ***$P < 0.001$ by one-way ANOVA with Bonferroni's post test. Specific $P$ values are provided in the Source Data file. Scale bars: **b** 20 μm; **d**, **e** 30 μm.

surfaces, the rate mineral apposition rate on trabecular, periosteal, and endocortical bone surfaces, or the number of osteoclasts on bone surfaces (Fig. 2f–j). Moreover, the serum biochemical results revealed that the KK-Ay and HFD + STZ mice had significantly lower serum procollagen type 1 N-terminal propeptide (P1NP) levels and higher C-terminal telopeptide of type 1 collagen (CTX-1) and sclerostin levels than the non-diabetic mice (Fig. S1).

### The osteocytic Ca²⁺ oscillatory response to mechanical loading is compromised in T2D both in vitro and in situ

Using advanced multi-scale cellular Ca²⁺ imaging techniques, we proceeded to examine the real-time intracellular Ca²⁺ oscillatory response to mechanical loading in the bone cells of T2D model animals. The system used in this study, based on a unique synchronized loading/imaging technique, has the capacity to record real-time Ca²⁺ dynamics in situ in the osteocyte networks of living murine long bones simultaneously subjected to cyclic compressive loading (Fig. 3a, Fig. S2). We found that in response to dynamic loading, the tibial osteocytes in normal male non-diabetic mice exhibited Ca²⁺ oscillations characterized by robust and repetitive Ca²⁺ spikes, whereas in both male KK-Ay and HFD + STZ mice, osteocytes produced only a few weak Ca²⁺ spikes in response to mechanical loading (Fig. 3b). Furthermore, statistical analyses revealed significant reductions in the percentage of Ca²⁺-responsive cells, Ca²⁺ spike number, and Ca²⁺ spike intensity, and a significant increase in the spike relaxation time in the osteocytes of KK-Ay and HFD + STZ mouse skeletons (Fig. 3b). These in situ studies were complemented by in vitro investigations, in which we examined real-time Ca²⁺ signaling in bone cells exposed to fluid shear stress (FSS) in a laminar flow chamber (Fig. 3c). Consistent with the findings of Ca²⁺ signaling in situ, we detected Ca²⁺ oscillations with repetitive robust Ca²⁺ spikes in response to steady FSS stimulation in both normal and mannitol-treated (the osmolarity control) MLO-Y4 osteocytic cells in vitro (Fig. 3d). In contrast, exposure to an HGHF environment induced a significant attenuation of the Ca²⁺ oscillatory response to steady FSS stimulation in MLO-Y4 cells, as characterized by a markedly reduced percentage of responsive cells, Ca²⁺ lower spike numbers and magnitudes, and a prolonged Ca²⁺ spike relaxation time (Fig. 3d). HGHF-mediated attenuation of Ca²⁺ dynamics in MLO-Y4 cells were confirmed in response to oscillatory FSS stimulation (Fig. S4). Moreover, we found that compared with osteocytes, primary osteoblasts were characterized by weaker Ca²⁺ oscillatory dynamics (releasing only one or two Ca²⁺ spikes) in response to FSS stimulation (Fig. 3e). Contrastingly, the Ca²⁺ profiles of HGHF-treated primary osteoblasts showed a minor decrease in the number of Ca²⁺ spikes and no significant difference in the Ca²⁺ intensity compared with those observed in normal or mannitol-treated (the osmolarity control) primary osteoblasts (Fig. 3e). Our findings thus provide compelling evidence to indicate that the T2D-induced reduction in bone mechano-responsiveness is primarily associated with the attenuated Ca²⁺ oscillatory dynamics in osteocytes.

### T2D-induced specific reduction of SERCA2 expression mediates an attenuation of osteocyte mechano-responsiveness

We subsequently sought to elucidate the mechanisms whereby T2D promotes an attenuation of osteocyte mechano-responsiveness. Considering that the endoplasmic reticulum (ER) is the main intracellular Ca²⁺ store, we depleted Ca²⁺ in the ER by treating osteocytes with thapsigargin or cyclopiazonic acid (CPA), two antagonists of SERCA (an intracellular pump located in the ER responsible for transporting Ca²⁺ back into the ER after Ca²⁺ release[30]). We found that treatment with thapsigargin or CPA promoted a significant suppression of osteocyte Ca²⁺ oscillations in MLO-Y4 cells in vitro in response to steady FSS stimulation and also in intact long bones in situ in response to compressive loading (Fig. 4a, b). After treatment with thapsigargin, FSS-induced increase in the expression of β-catenin and OPG/RANKL and decrease in DKK1 were abolished in MLO-Y4 cells (Fig. S5). Then, we treated osteocytes with the antagonists those blocked the upstream signaling molecules mediating the Ca²⁺ release from the ER. We found that treatment with the P₂R antagonist suramin, the PLC antagonist neomycin, or the IP₃R antagonist 2-aminoethoxydiphenyl borate (2-APB) resulted in a significant inhibition of FSS-induced Ca²⁺ oscillations in normal MLO-Y4 cells and totally abolished the FSS-induced Ca²⁺ spikes in HGHF-treated MLO-Y4 cells (Fig. S5). Moreover, we found that exposure to an HGHF environment is associated with significant reductions in the gene and protein expression of SERCA2 in MLO-Y4 osteocyte cells, although not in the expression of SERCA1 or SERCA3 (Fig. 4c, d). Our qRT-PCR and immunohistochemical results revealed that the expression of SERCA2 in tibial osteocytes in vivo, although neither that of SERCA1 nor SERCA3, was significantly reduced in both male KK-Ay and HFD + STZ mice compared with that in the non-diabetic mice (Fig. 4e, f). However, no significant difference was observed in the *ATP2A2* (the gene encoding the SERCA2 protein) mRNA stability or SERCA2 protein stability between normal and HGHF-treated MLO-Y4 cells (Fig. S6). Furthermore, we found no significant difference in the protein expression of P₂Y isoforms (P₂Y₂, P₂Y₄, and P₂Y₁₂, three major functional isoforms expressed in osteocytes[31]), PLCβ1 (P₂Y-downstream effector isoform), or IP₃R between normal and HGHF-treated MLO-Y4 cells (Fig. S6). Moreover, we transfected MLO-Y4 cells with siRNAs targeting *ATP2A2*, and the knockdown efficacy is shown in Fig. S7. In the *ATP2A2*-silenced MLO-Y4 cells, we observed attenuated Ca²⁺ oscillatory signaling profiles in response to steady FSS stimulation, characterized by a significantly lower Ca²⁺ spike number and intensity and a longer relaxation time than those observed in normal MLO-Y4 cells (Fig. 4g), which were similar to the effects of broad-spectrum SERCA antagonists in osteocytes (Fig. 4a, b). In addition, we observed an almost complete disappearance of oscillatory FSS-induced osteocyte Ca²⁺ oscillations in MLO-Y4 cells in vitro following treatment with thapsigargin, CPA, or *ATP2A2* siRNA (Fig. S8). Collectively, our findings highlight the importance of a T2D-induced specific reductions in SERCA2 expression in the attenuated mechano-responsiveness of osteocytes.

### Increasing SERCA2 expression in T2D skeletons enhances biological activities and Ca²⁺ oscillatory responses to mechanical loading in osteocytes, although not in osteoblasts

We proceeded to further examine whether enhancing the expression of SERCA2 using an SERCA2 agonist could rescue the osteocyte mechanoresponse in the skeletons of T2D mice. For this purpose, we selected ISTA, an agonist known to selectively stimulate SERCA2

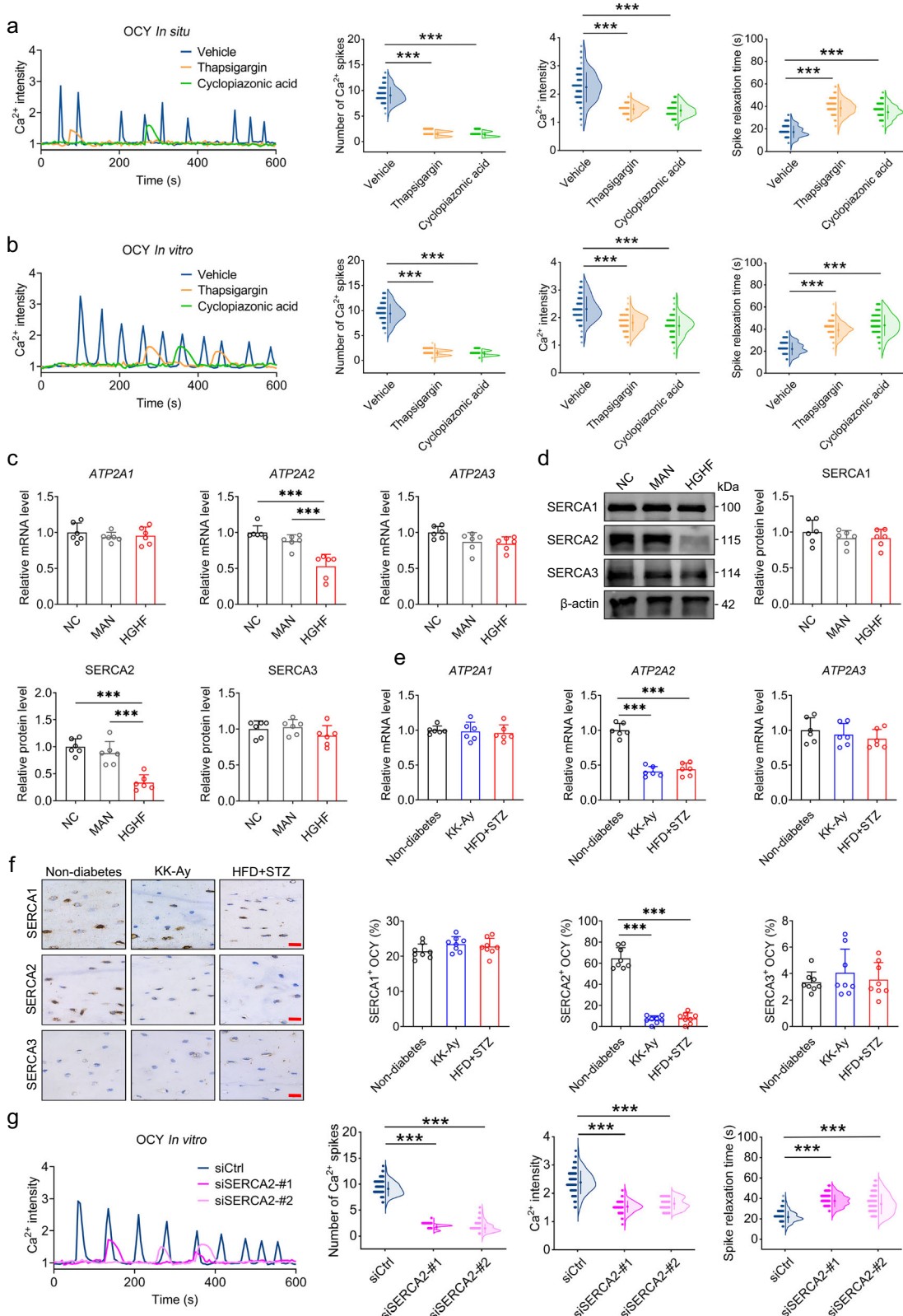

activity, which has been used to enhance cardiac function in patients with heart diseases[32]. We found that in situ, following ISTA treatment, tibial osteocytes in both male KK-Ay and HFD + STZ mice exhibited $Ca^{2+}$ oscillations characterized by multiple robust $Ca^{2+}$ spikes in response to cyclic compressive loading (Fig. 5a, b), which were similar to the $Ca^{2+}$ profiles in non-diabetic mice. Statistical analyses revealed that in both KK-Ay and HFD + STZ mice, ISTA treatment resulted in a

significant increase in the $Ca^{2+}$ spike number and $Ca^{2+}$ spike intensity, and a significant reduction in the spike relaxation time (Fig. 5a, b). Furthermore, the ISTA-mediated restoration in $Ca^{2+}$ oscillatory dynamics was also observed in HGHF-exposed MLO-Y4 cells in vitro in response to steady FSS stimulation (Fig. 5c). Contrastingly, treatment with this agonist had no observable effects on intracellular $Ca^{2+}$ dynamics in response to FSS stimulation in HGHF-exposed primary

**Fig. 4 | T2D-induced specific reduction of SERCA2 expression mediates an attenuation of osteocyte mechano-responsiveness. a** Real-time intracellular Ca²⁺ signaling of osteocytes in situ in mouse tibiae treated with DMSO (vehicle control) and two antagonists of SERCA, i.e., thapsigargin (1 μM) and cyclopiazonic acid (CPA; 5 μM), in response to subsequent cyclic mechanical loading. **b** Intracellular Ca²⁺ signaling in MLO-Y4 osteocytic cells in vitro treated with DMSO, thapsigargin, and CPA in response to subsequent steady FSS stimulation at 2 Pa. **c** qRT-PCR analyses of the *ATP2A1*, *ATP2A2* and *ATP2A3* gene expression in MLO-Y4 osteocytic cells treated with the normal medium (NC), mannitol-supplemented medium (MAN, osmotic pressure-matched control), and high-glucose and high-fat (HGHF)-supplemented medium. **d** Western blotting analyses of the SERCA1, SERCA2, and SERCA3 protein expression in normal, mannitol-treated, and HGHF-treated MLO-Y4 osteocytic cells. **e** qRT-PCR analyses of the *ATP2A1*, *ATP2A2* and *ATP2A3* gene expression in male non-diabetic, KK-Ay, and HFD + STZ mice. **f** Immunohistochemical staining of SERCA1, SERCA2, and SERCA3 in osteocytes in the mouse tibiae of the non-diabetes, KK-Ay, and HFD + STZ groups. **g** Intracellular Ca²⁺ signaling in MLO-Y4 osteocytic cells in vitro treated with siCtrl and siSERCA2 in response to steady FSS stimulation at 2 Pa. Graphs represent mean ± SD (**a**, **b**, **g**: *n* = 120 cells per group; **c–e** *n* = 6 biologically independent replicates; **f**: *n* = 8 mice per group). \*\*\**P* < 0.001 by one-way ANOVA with Bonferroni's post test. Specific *P* values are provided in the Source Data file. Scale bars: **f** 20 μm.

osteoblasts (Fig. 5d). Similarly, ISTA induced a significant enhancement of Ca²⁺ dynamics in HGHF-treated MLO-Y4 cells in response to oscillatory FSS stimulation (Fig. S9). T2D-induced suppression in SERCA2 expression and Ca²⁺ oscillatory dynamics in osteocytes was confirmed in 3-month-old female KK-Ay mice (Fig. S10), and ISTA enhanced osteocytic Ca²⁺ oscillatory responses to mechanical loading in these female mice (Fig. S10).

We subsequently investigated changes in the biological activities of HGHF-exposed osteocytes and osteoblasts in response to FSS following the ISTA treatment. Annexin V-FITC/PI apoptosis and CCK-8 assays revealed that FSS stimulation induced a significant reduction in cell apoptosis and a significant increase in cell viability in normal MLO-Y4 cells, although not in HGHF-exposed cells (Fig. 5e and Fig. S11). Following ISTA treatment, HGHF-exposed osteocytes exhibited a significant reduction in cell viability in response to FSS stimulation (Fig. 5e and Fig. S11). Moreover, in response to FSS stimulation, we detected a significant increase in the gene and protein expression of β-catenin and OPG and a significant reduction in RANKL and DKK1 expression in normal MLO-Y4 cells, although not in HGHF-treated MLO-Y4 cells (Fig. 5f and Fig. S11). Having administered ISTA to HGHF-exposed MLO-Y4 cells, we found the expression of β-catenin and OPG to be significantly increased and that of RANKL and DKK1 to be significantly reduced in response to FSS stimulation (Fig. 5f and Fig. S11). Moreover, FSS stimulation promoted a significant upregulation of the gene and protein expression of Col1a1, Osx, and Runx2, as well as inducing ALP activity and the formation of mineralized nodules in normal and HGHF-exposed primary osteoblasts (Fig. 5g–j and Fig. S11). Contrastingly, FSS stimulation had no observable effects on osteogenic differentiation or mineralization in HGHF-exposed osteoblasts or HGHF-exposed osteoblasts treated with ISTA (Fig. 5g–j and Fig. S11). Collectively, these findings clearly indicate that inducing an increase of SERCA2 expression would be an effective approach to enhance osteocyte mechanoresponse in T2D skeletons.

### Treatment with the SERCA2 agonist ISTA enhances bone mechano-responsiveness in KK-Ay mice

We went on to examine whether treatment with the SERCA2 agonist ISTA would promote bone mechano-responsiveness in genetically spontaneous T2D mice (KK-Ay mice) in response to a cyclic compressive loading of 1200 cycles/day for 2 weeks (Fig. 6a). Administration of ISTA appeared to exert no obvious effect on either the horizontal or vertical activity of KK-Ay mice (Fig. S12). The results of immunohistochemical staining confirmed an increase in the expression of SERCA2 following ISTA treatment (Fig. S13), and micro-CT results revealed a significant increase in tibial trabecular bone mass and thickness and a significant reduction in cortical bone porosity in the loaded tibiae of ISTA-treated KK-Ay mice, although not in those of non-treated KK-Ay mice (Fig. 6b–d). Furthermore, the results of biomechanical three-point bending and indentation tests revealed significant improvements in whole-bone mechanical properties and intrinsic material properties of cortical and trabecular bone in the tibiae of ISTA-treated KK-Ay mice subjected to exogenous cyclic compressive loading (Fig. 6e and Fig. S13). In the context of the ISTA

treatment, we established that exogenous cyclic loading had the effect of significantly suppressing osteocytic sclerostin and RANKL expression in the tibiae of KK-Ay mice, and increased the expression of OPG (Fig. 6f–h). In addition, KK-Ay mice that had received ISTA treatment were found to be characterized by a significant reduction in the number of osteoclasts on tibial trabecular bone surfaces and a larger number of osteoblasts on bone surfaces, along with an enhanced rate of trabecular bone mineral apposition, in response to exogenous cyclic loading (Fig. 6i–k). Moreover, we found that compared with mice in the KK-Ay group, the KK-Ay+ISTA mice exhibited a significant increase in canalicular length and density and a significant reduction in the apoptosis of osteocytes in response to exogenous cyclic loading (Fig. S13). ISTA treatment also induced a significant increase in serum P1NP and a decrease in serum CTX-1 and sclerostin concentrations in KK-Ay mice (Fig. S13).

### ISTA treatment enhances bone mechano-responsiveness in HFD + STZ-induced T2D mice via the osteocyte-mediated regulation of osteoblasts and osteoclasts

In further studies investigating the effects of ISTA treatment, we examined bone mechano-responsiveness to a 2-week period of exogenous cyclic compressive loading (1200 cycles/day) in experimentally-induced (HFD + STZ) T2D mice (Fig. 7a). Open field behavior testing revealed no significant difference in horizontal or vertical activities between ISTA-treated and non-treated HFD + STZ mice (Fig. S12). However, immunohistochemical staining results indicated a significant upregulation in the expression of SERCA2 following ISTA treatment (Fig. S14). Micro-CT results revealed that in response to exogenous cyclic loading, mice in the HFD + STZ + ISTA group were characterized by significantly higher cortical bone thickness and trabecular bone mass and thickness and lower cortical porosity than those in the HFD + STZ group (Fig. 7b, c and Fig. S14). Consistently, biomechanical analyses provided evidence to indicate that ISTA induced a significant increase in whole-bone mechanical strength and pronounced improvements in the intrinsic material properties of cortical and trabecular bone in the tibiae of HFD + STZ mice subjected to exogenous cyclic loading (Fig. 7d and Fig. S14). In addition, we detected loading-induced increase in canalicular length and density and reduction in the apoptosis of osteocytes in the tibiae of ISTA-treated HFD + STZ mice, although not in non-treated HFD + STZ mice (Fig. S14). Moreover, in response to exogenous loading, the osteocytes of ISTA-treated HFD + STZ mice were found to be characterized by significantly lower RANKL expression and higher expression of OPG than the non-treated HFD + STZ mice (Fig. 7e, f and Fig. S14). ISTA treatment also promoted a marked increase in the number of osteoblasts on tibial bone surfaces in HFD + STZ mice subjected to exogenous cyclic loading (Fig. 7g). HFD + STZ mice that had received ISTA treatment exhibited a significant increase in serum P1NP and a significant reduction in serum CTX-1 and sclerostin levels (Fig. S14). We subsequently investigated the regulatory role of osteocytes in ISTA-induced bone mechanical sensitization in T2D mice. Following treatment with ISTA, HGHF-exposed MLO-Y4 cells were subjected to FSS stimulation, and thereafter the conditioned medium was added into

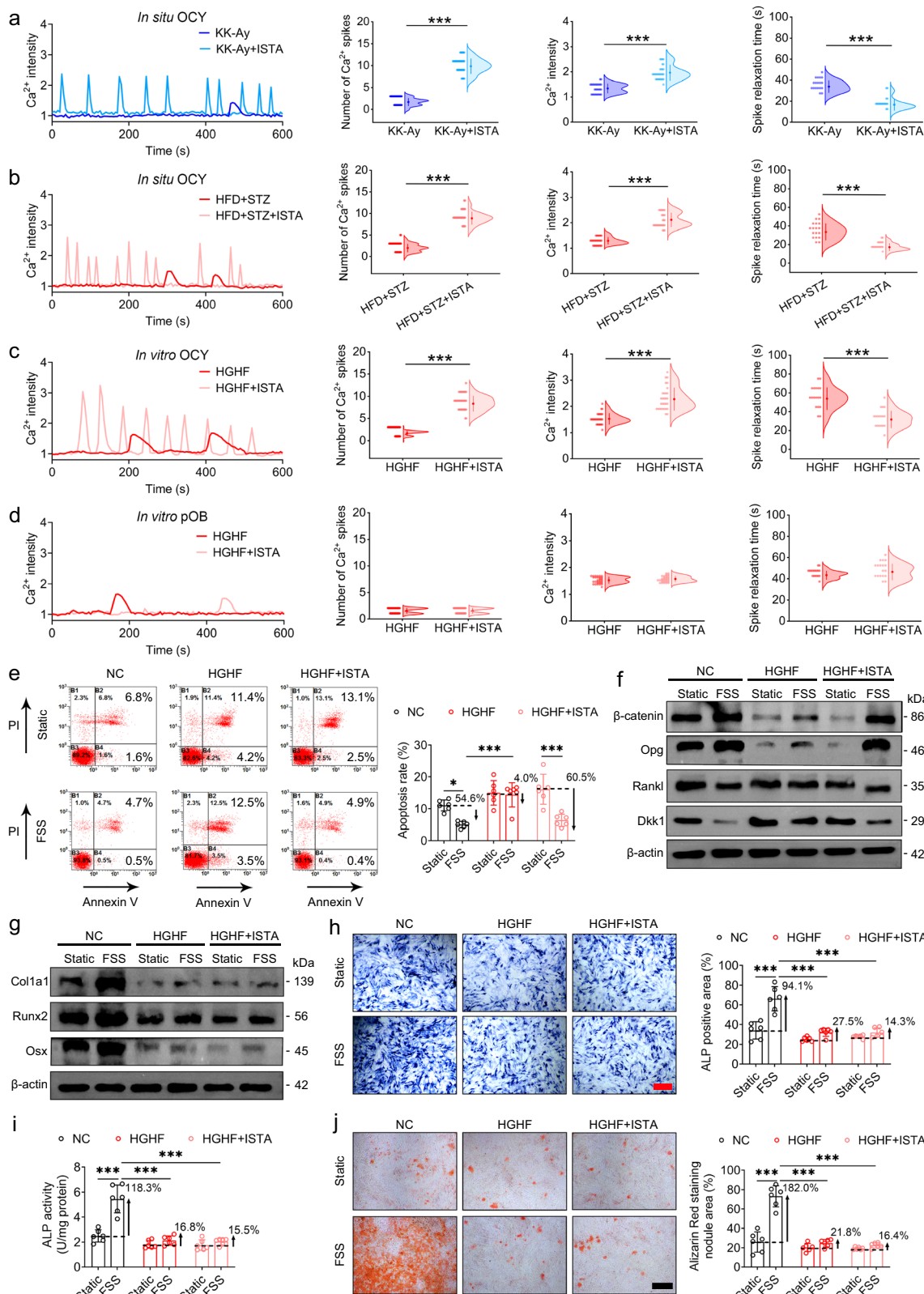

HGHF-exposed osteoclasts and osteoblasts (Fig. 7h). We accordingly found that compared with conditioned medium collected from MLO-Y4 cells in the HGHF + FSS group, that collected from MLO-Y4 cells in the HGHF + ISTA + FSS group promoted a significant enhancement of osteoblast differentiation and mineralization, as evidenced by the upregulated gene and protein expression of osteogenic markers (Col1a1, Runx2, and Osx), along with increases in ALP activity, the

number of ALP-positive cells, and the area of calcified nodules (Fig. 7i–k and Fig. S15). Moreover, the conditioned medium from osteocytes in the HGHF + ISTA + FSS group resulted in a significant reduction in osteoclast differentiation and function compared with that collected from osteocytes in the HGHF + FSS group, as revealed by the findings of osteoclast-related marker expression assays (Fig. 7l, m and Fig. S15). Our findings thus provide comprehensive evidence to

**Fig. 5 | Increasing SERCA2 expression in T2D skeletons enhances biological activities and Ca²⁺ oscillatory responses to mechanical loading in osteocytes, although not in osteoblasts. a** Real-time intracellular Ca²⁺ signaling of osteocytes in situ in the tibiae of male KK-Ay mice treated with the SERCA2 agonist istaroxime (ISTA, 1 μM) in response to axial cyclic compressive loading. **b** Intracellular Ca²⁺ dynamics of osteocytes in situ in the tibiae of high-fat diet/streptozotocin-treated (HFD + STZ) male mice incubated with ISTA in response to cyclic compressive loading. **c** Intracellular Ca²⁺ dynamics of MLO-Y4 osteocytic cells in vitro treated with high glucose and high-fat (HGHF) and the SERCA2 agonist ISTA (1 μM) in response to steady FSS stimulation (2 Pa). **d** Intracellular Ca²⁺ dynamics of primary osteoblasts in vitro treated with HGHF and ISTA in response to steady FSS

stimulation. **e** The Annexin V-FITC/PI apoptosis assays in MLO-Y4 cells treated with HGHF and ISTA in response to FSS stimulation (2 Pa). **f** Western blotting assays in MLO-Y4 cells treated with HGHF and ISTA in response to FSS, including β-catenin, OPG, RANKL, and DKK1. **g** Western blotting assays in primary osteoblasts treated with HGHF and ISTA in response to FSS, including Col1a1, Osx, and Runx2. **h**–**j** The ALP activity, ALP staining, and Alizarin red staining in primary osteoblasts treated with HGHF and ISTA in response to FSS stimulation. Graphs represent mean ± SD (**a**–**d** $n = 120$ cells per group; **e**–**j** $n = 6$ biologically independent replicates). **a**–**d** ***$P < 0.001$ by two-tailed Student's $t$ test. **e**–**j** *$P < 0.05$ and ***$P < 0.001$ by two-way ANOVA with Bonferroni's post test. Specific $P$ values are provided in the Source Data file. Scale bars: **h**, **j** 50 μm.

indicate that by increasing the expression of SERCA2, ISTA treatment can promote a significant improvement in bone mechano-responsiveness of T2D mice via the osteocyte-mediated regulation of osteoblasts and osteoclasts.

## Specific overexpression of SERCA2 in osteocytes suppresses the T2D-induced deterioration in bone mechano-responsiveness

To further investigate the role of osteocytic SERCA2 in the T2D-mediated deterioration of bone mechano-responsiveness, we generated a conditional osteocyte SERCA2 knock-in mouse model (*SERCA2* cKI) by crossing SERCA2^flox/flox mice with DMP1-Cre mice (Fig. S16). The efficiency of SERCA2 overexpression in osteocytes of this mouse model was confirmed based on PCR and immunohistochemical staining analyses (Fig. S16). The SERCA2 cKI mice were injected with low-dose STZ for 5 days combined with HFD feeding, and thereafter were subjected to 2 weeks of exogenous cyclic loading (Fig. 8a). Micro-CT results revealed a loading-mediated significant increase in tibial cortical bone thickness and trabecular bone mass and thickness and a reduction in cortical bone porosity in the *SERCA2* cKI mice treated with HFD + STZ, although not in the WT mice treated with HFD + STZ (Fig. 8b–d). Moreover, biomechanical three-point bending and nanoindentation tests revealed that specific overexpression of SERCA2 in osteocytes promoted significant improvements in both whole-bone mechanical strength and the intrinsic local material properties of the tibiae of HFD + STZ mice in response to cyclic compressive loading (Fig. 8e, f and Fig. S17). In addition, we detected exogenous cyclic loading-induced reductions in RANKL and sclerostin protein expression and cellular apoptosis in osteocytes and an increase in the expression of osteocytic OPG and canalicular length and density in the *SERCA2* cKI mice treated with HFD + STZ, although not in the similarly treated WT mice (Fig. 8g, h and Fig. S17). Compared with the HFD + STZ-treated WT mice, *SERCA2* cKI mice subjected to HFD + STZ were also found to be characterized by significantly higher rates of trabecular bone formation, fewer osteoclasts and more osteoblasts on bone surfaces in response to exogenous cyclic loading (Fig. 8i–k). HFD + STZ-treated *SERCA2* cKI mice exhibited a significant increase in serum P1NP and a significant decrease in serum CTX-1 and sclerostin concentrations as compared with the HFD + STZ-treated WT mice (Fig. S17). However, we detected no significant differences between the WT and *SERCA2* cKI mice with respect to body weight, food intake, or horizontal/vertical activity (Fig. S17). Our findings based on analyses of the SERCA2 knock-in model indicate that specific overexpression of SERCA2 in osteocytes contributes to stemming the deterioration of bone mechano-responsiveness induced by T2D.

## The nuclear transcription factor PPARα mediates T2D-induced specific reduction in the expression of osteocytic SERCA2 pump

We proceeded to further examine the mechanism whereby T2D-induced specific reduction of SERCA2 expression in osteocytes. KEGG pathway enrichment analyses based on RNA sequencing (RNA-seq) showed that the PPAR signaling pathway ranked first in the top 10 enriched KEGG pathways, revealing that the PPAR-related events were strongly affected by HGHF (Fig. 9a). PPARs (containing three different

isotypes, including PPARα, PPARβ/δ, and PPARγ) are members of the steroid hormone receptor superfamily that serve as metabolic sensors and central regulators of glucose and fat homeostasis[33]. Our gene-set enrichment analysis (GSEA) revealed a significant enrichment of the signaling events associated with PPARα and PPAR_DR1_Q2 (a binding motif of the PPARα isoform) in HGHF-exposed MLO-Y4 cells (Fig. 9b, c), which indicated a possible decrease of the PPARα activity following HGHF exposure. Moreover, western blotting results confirmed HGHF-induced significant reduction in the PPARα protein expression and significant increase in p-PPARα in MLO-Y4 cells (Fig. 9d). The immunohistochemical results indicate that both KK-Ay and HFD + STZ mice exhibited significantly lower expression of PPARα in tibial osteocytes than that in non-diabetic mice (Fig. 9e). Moreover, compared with normal MLO-Y4 cells, a significant decrease in the gene and protein expression of SERCA2 expression was observed in MLO-Y4 cells treated with the PPARα antagonist (MK886 or GW6471), although not in cells treated with the PPARβ/δ antagonist (GSK0660 or GSK3787) or PPARγ antagonist (T0070907 or GW9662) (Fig. 9f, Fig. S18). Consistently, after treatment with the PPARα agonist fenofibrate or GW7647, HGHF-induced decrease in the gene and protein expression of SERCA2 was significantly suppressed (Fig. 9g, Fig. S18). However, treatment with the PPARβ/δ or PPARγ agonists had no significant effects on the gene and protein expression of SERCA2 in HGHF-treated MLO-Y4 cells (Fig. 9g). Moreover, PPARα was found to have potential binding sites that could recognize the promoter sequence of *ATP2A2* according to transcription factor prediction (Fig. 9h). Then, we truncated the predicted binding sites (with promotor sequence −1300 to +1 bp) of PPARα recognizing *ATP2A2* into three successively smaller fragments (Fig. 9h), and found that the promoter region (−620 to −608 bp) instead of the promoter region (−1283 to −1277 bp) was the binding site recognized by PPARα (Fig. 9h). The promotor activities of all shaved fragments are shown in Fig. 9i. Furthermore, the luciferase activity was found to be significantly decreased in HGHF-exposed MLO-Y4 cells, and suppressing the PPARα expression resulted in no significant difference in the luciferase activity between the NC and HGHF groups (Fig. 9j). The chromatin immunoprecipitation (ChIP) assays revealed that the enrichment of PPARα on *ATP2A2* promoter was significantly decreased in HGHF-treated MLO-Y4 cells (Fig. 9k). Then, we performed electrophoretic mobility shift assays (EMSA) to provide further evidence that PPARα bound to the *ATP2A2* promoter region (−620 to −608 bp) rather than the region (−1283 to −1277 bp) (Fig. 9l). Furthermore, the EMSA results showed that the binding activity of PPARα to the *ATP2A2* promoter was significantly decreased under the HGHF condition (Fig. 9l). MLO-Y4 cells with lentiviral over-expression of PPARα that had exposed to HGHF exhibited a significant enhancement of Ca²⁺ oscillatory response to FSS stimulation compared with HGHF-exposed cells infected with LvCtrl, along with significant increases in the gene and protein expression of β-catenin and OPG and significant reductions in RANKL and DKK1 expression (Fig. 9m, n and Fig. S18). Moreover, FSS-induced Ca²⁺ oscillation, increased β-catenin and OPG expression, and decreased RANKL and DKK1 expression were suppressed in MLO-Y4 cells following lentiviral knockdown of PPARα (Fig. 9o, p and Fig. S18).

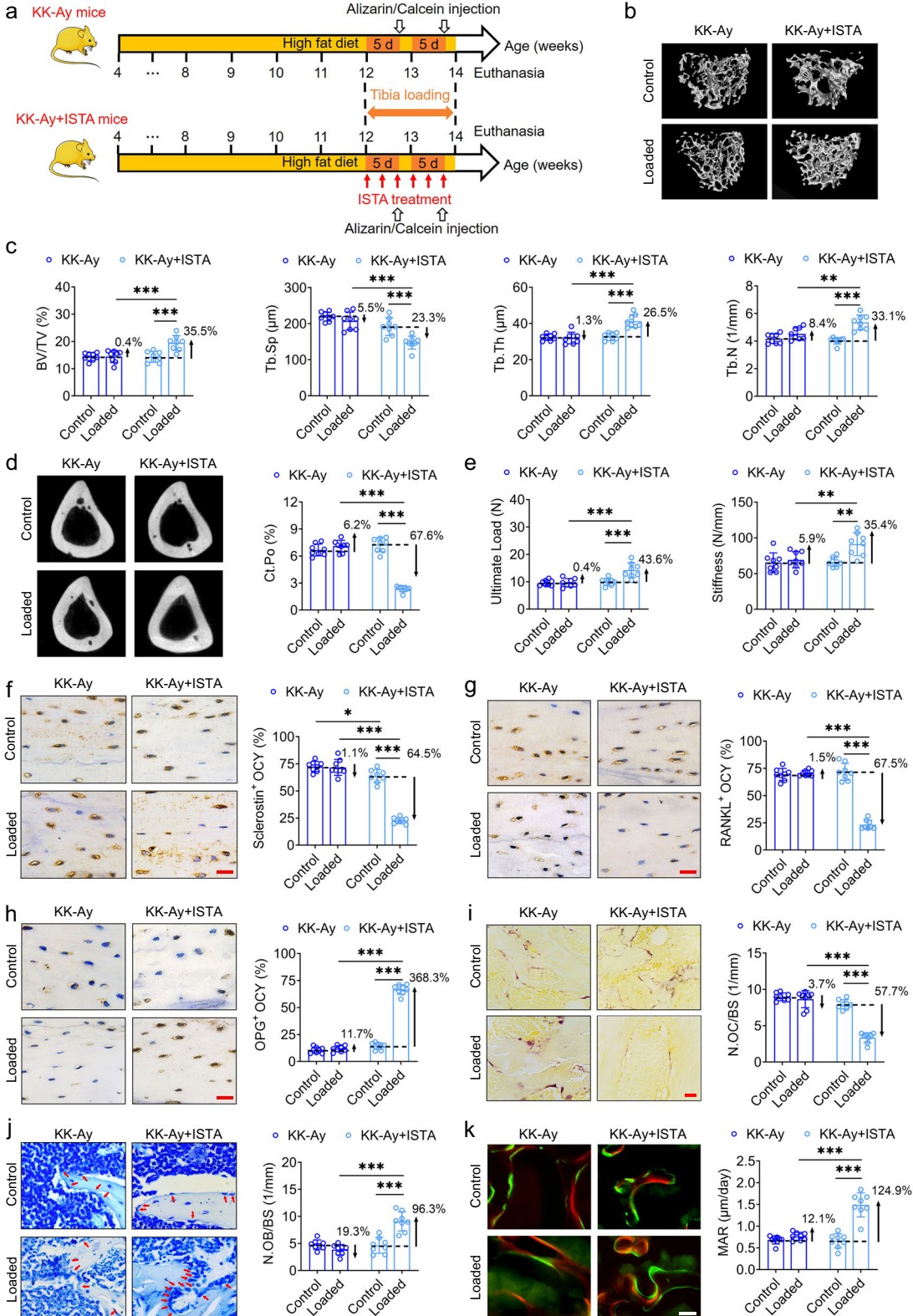

## Discussion

Given the ongoing trend of global aging, T2D-related fragility fractures pose an increasingly difficult medical challenge[34], and in this regard, considering the significance of mechanical loading in the maintenance of bone integrity and homeostasis, it is necessary to characterize the mechano-responsiveness of bone in patients with T2D. Herein, we provide direct evidence that exogenous cyclic loading-mediated improvements in bone architecture and strength are compromised

in two T2D animal models (KK-Ay and HFD + STZ mice), which we primarily identified as being associated with the altered Ca²⁺ oscillatory dynamics of osteocytes, although not those of osteoblasts. Furthermore, we found that the T2D-induced reduction of bone mechano-responsiveness was attributed to PPARα-mediated specific reduction in the expression of osteocytic SERCA2. Treatment with the SERCA2 agonist ISTA or endogenous specific overexpression of SERCA2 in osteocytes was demonstrated to reverse the deterioration in the

**Fig. 6 | Treatment with the SERCA2 agonist ISTA improves bone mechano-responsiveness in genetically spontaneous T2D KK-Ay mice. a** The experimental protocol of the ISTA treatment (intravenous infusion of 73 µg/kg body weight per min for 15 min every other day with three times per week from the first day of exogenous cyclic loading), and subsequent application of cyclic compressive loading (1200 cycles/day) on the tibiae in KK-Ay mice for 2 weeks.
**b–d** Representative micro-CT images showing proximal tibial trabecular bone microstructure and cortical bone porosity in KK-Ay mice, and the corresponding statistical results. **e** Three-point bending testing to analyze whole-bone mechanical properties in KK-Ay mice. **f–h** Immunohistochemical staining showing the protein expression of sclerostin, RANKL, and OPG in osteocytes in tibial cortical bone matrix of KK-Ay mice. **i** Representative TRAP staining to label osteoclasts on tibial bone surfaces in KK-Ay mice, and the corresponding statistical result. **j** Toluidine blue staining to label osteoblasts on tibial trabecular bone surface in KK-Ay mice. **k** Representative calcein and alizarin red double labeling of tibial trabecular bone surfaces in KK-Ay mice, and the corresponding statistical result. Graphs represent mean ± SD ($n = 8$ mice per group). *$P < 0.05$, **$P < 0.01$, and ***$P < 0.001$ by two-way ANOVA with Bonferroni's post test. Specific $P$ values are provided in the Source Data file. Scale bars: **f–h, j** 20 µm; **i** 30 µm; **k** 50 µm.

mechanoresponse of T2D skeletons by rescuing osteocyte $Ca^{2+}$ dynamics and the osteocyte-mediated regulation of osteoblasts and osteoclasts. These findings could provide the basis of a promising alternative therapeutic approach for ameliorating T2D-induced bone fragility from the perspective of regulating osteocyte mechano-responsiveness.

Regular weight-bearing exercise is beneficial for the maintenance of bone health, and emerging evidence suggests that a reduction in bone mechano-responsiveness negatively influences bone status[14,35]. On the basis of a unilaterally constrained hindlimb loading model allied to a controllable physiological loading scheme[36], we studied bone adaptation in response to exogenous cyclic loading in normal and T2D animals. We herein utilized KK-Ay and HFD + STZ mice, two animal models proven to be effect in studying T2D-related complications[37], to study bone mechano-responsiveness. We observed a significant decrease in bone formation and increase in bone resorption in these mice, which were in line with previous findings[38,39]. In humans, bone formation was shown to be reduced in T2D, while bone resorption was less consistently reported (most researchers observed a decrease in bone resorption in patients with T2D, and others revealed a significant increase)[40,41]. In response to exogenous cyclic compressive loading, we observed significant improvements in bone architecture, strength, material properties, and formation in non-diabetic mice, which validates the efficacy of the adopted loading protocol. In contrast, the same mechanical strains induced by axial tibial compression in KK-Ay and HFD + STZ mice failed to compensate for the T2D-induced deteriorations in trabecular and cortical bone architecture or mechanical properties. Therefore, this study provides strong evidence to identify the attenuated adaptive response to exogenous cyclic loading in T2D skeletons.

The proposed mechanisms whereby T2D impairs the mechanoresponse of bone is schematically represented in Fig. 10. On the basis of our unique multi-scale $Ca^{2+}$ imaging platform, we observed load-induced intracellular $Ca^{2+}$ oscillations in osteocytes, characterized by multiple robust spikes, both in vitro and in situ, which bear similarities to the repetitive firing of neuronal cells[42]. We found that these $Ca^{2+}$ oscillations are primarily attributable to the $Ca^{2+}$ release from ER stores through $P_2R$-mediated activation of PLC-IP$_3$ receptor pathway followed by the reuptake of $Ca^{2+}$ into the ER via the SERCA pumps. In contrast, in response to mechanical loading, osteocytes in the skeletons of T2D mice exhibited fewer and weaker spikes, together with a prolongation of $Ca^{2+}$ reuptake. Compared with constant $Ca^{2+}$ signals, oscillating $Ca^{2+}$ signals have been established to be considerably more efficient in promoting the enhancement of gene transcription, cellular contraction, and cytokine efflux[24,43,44], which is consistent with our findings that the load-mediated regulation of osteocytic cytokine production is almost totally abolished in response to the suppression of $Ca^{2+}$ dynamics. However, we observed no appreciable differences in the $Ca^{2+}$ profiles of normal and T2D osteocytes. Our findings accordingly indicate that in terms of mechano-responsiveness, osteocytes are notably more vulnerable to an HGHF environment than osteoblasts, and thus modulating osteocyte $Ca^{2+}$ dynamics could represent an effective approach for enhancing the mechanoresponse of bone in individuals with T2D.

Our results revealed a specific reduction in the expression of SERCA2 in HGHF-treated osteocytes. Similarly, several previous studies have also revealed the high vulnerability of SERCA2 to T2D in myocardial cells, vascular smooth muscle cells, and endothelial cells[45,46]. As the most widespread of the multiple SERCA isoforms, SERCA2 has been established to be essential for the maintenance of $Ca^{2+}$ homeostasis by mediating the recycling of cytosolic $Ca^{2+}$ into the ER lumen[47]. Accumulating evidence indicates the involvement of SERCA2-dependent $Ca^{2+}$ dysregulation in the pathogenesis of a range of cardiovascular and neurodegenerative diseases[48,49]. In particular, the levels of SERCA2 are considered to be a key determinant of muscle contraction and heartbeat frequency and rhythmicity[50,51]. Intriguingly, in our previous study, we detected $Ca^{2+}$ transient-mediated cytoskeletal contractions in osteocytes in response to FSS[27], thereby revealing a close similarity to the behaviors observed in cardiac and muscle cells. In the present study, the deficiency of SERCA2 expression observed in T2D osteocytes contributes to low levels of $Ca^{2+}$ stored in the ER, which consequently incapacitates osteocytes from releasing/re-uptaking $Ca^{2+}$ vigorously during each $Ca^{2+}$ transient cycle, thereby resulting in attenuated $Ca^{2+}$ signatures in response to external mechanical loading.

ISTA is a positive inotropic agent used to enhance cardiac performance and hemodynamics by activating SERCA2, which optimizes cellular $Ca^{2+}$ handling[52,53]. In this study, we observed $Ca^{2+}$ profiles characterized by more robust and repetitive $Ca^{2+}$ transients in ISTA-treated T2D osteocytes than in untreated T2D osteocytes in response to mechanical loading, which we suspect to be primarily associated with an elevated $Ca^{2+}$ storage capacity and accelerated $Ca^{2+}$ cycling in the ER. In contrast to an absence of any observable direct effects of ISTA on osteoblastic $Ca^{2+}$ signatures or biological activities in response to FSS, we observed the osteocyte-mediated regulation of osteoblasts and osteoclasts following the exposure of ISTA-treated T2D osteocytes to FSS, which accordingly provides an explanation as to why ISTA treatment augments bone formation and inhibits bone resorption in T2D mice in response to exogenous cyclic loading. Importantly, we established that conditional osteocyte SERCA2 cKI mice are resistant to a T2D-induced reduction in bone mechano-responsiveness, which further emphasizes the importance of SERCA modulation in osteocytes. Our findings revealed that SERCA2 and SERCA2-mediated $Ca^{2+}$ dynamics could serve as key therapeutic targets for improving bone mechano-responsiveness in patients with T2D.

The nuclear transcription factor PPARα is a critical metabolic regulator and nutritional sensor, and plays a major role in the maintenance of glucose, lipid, and cholesterol homeostasis[54]. Abnormal alteration of PPARα is associated with the pathogenesis of many diseases, such as obesity, cardiovascular diseases, diabetes, and inflammation[55]. PPARα-mediated gene transcription involves a series of complex processes, such as interaction with heterodimeric partner, cofactor recruitment, and phosphorylation events[56]. Several studies have indicated that PPARα is abundantly expressed in musculoskeletons and shows a positive impact on bone mass and bone metabolism[57,58]. Our results not only revealed a specific reduction in PPARα activity following HGHF exposure, but also identified the transcriptional regulation of PPARα on *ATP2A2* and the positive

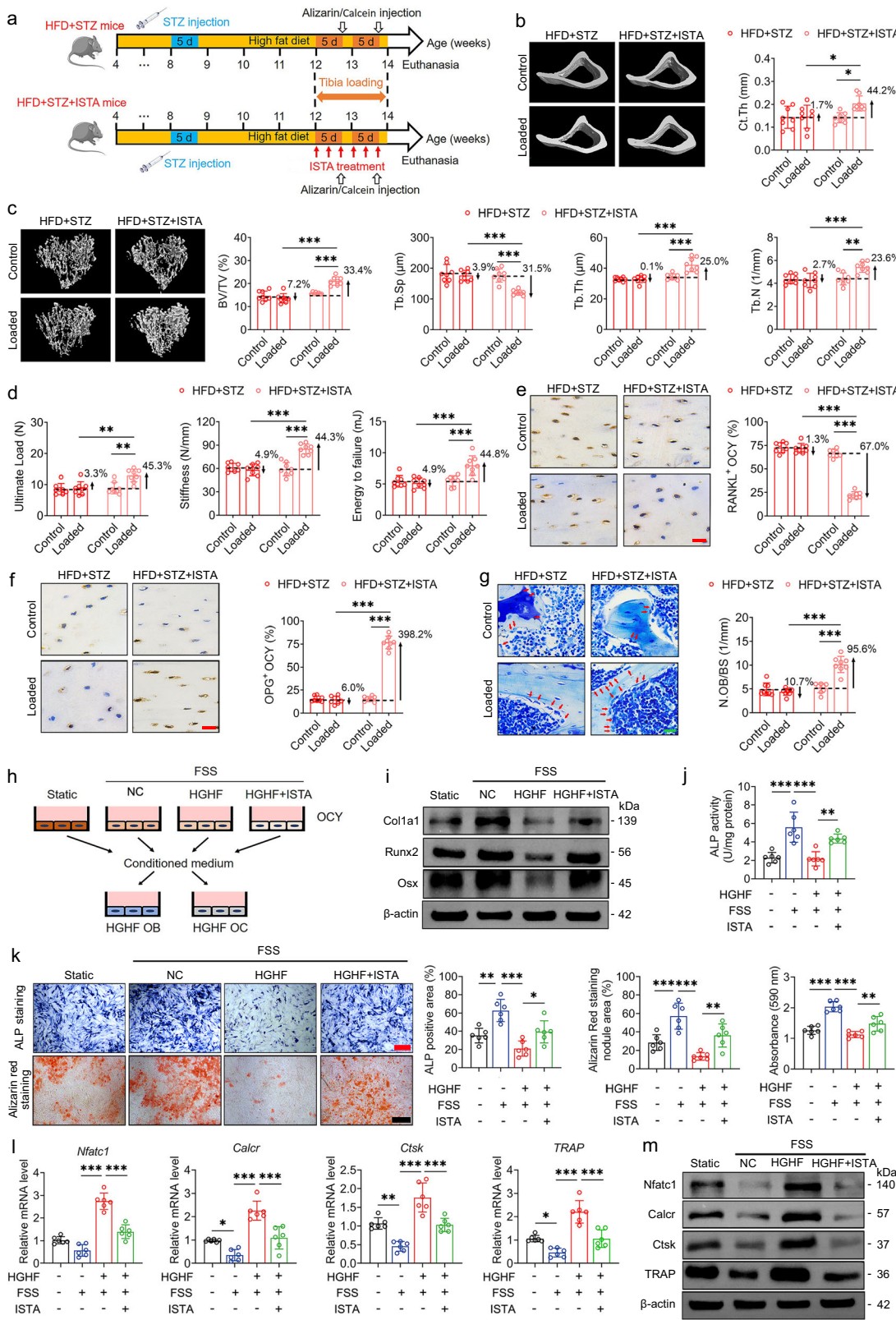

correlation between *ATP2A2* and PPARα (but not PPARβ/δ or PPARγ). Importantly, suppressing PPARα expression in normal osteocytes or rescuing PPARα in HGHF-exposed osteocytes bears similarities to the role of SERCA2 in the maintenance of bone mechano-responsiveness. Thus, our findings provide strong evidence to reveal that HGHF-induced specific reduction in osteocytic SERCA2 pump is primarily associated with the reduction in PPARα activity.

In conclusion, this study provides an interpretation of the mechanisms underlying T2D-induced reduction in bone mechano-responsiveness. This response was identified as being attributable to abnormal osteocytic Ca$^{2+}$ oscillatory dynamics, primarily induced by PPARα-mediated specific reduction in the expression of SERCA2. These findings provide strong evidence indicating the therapeutic potential of weight-bearing aerobic or resistance training exercise,

**Fig. 7 | ISTA treatment improves bone mechano-responsiveness in high-fat diet/streptozotocin-induced T2D mice via osteocytes-mediated regulation of osteoblasts and osteoclasts. a** The experimental protocol of ISTA treatment (intravenous infusion of 73 μg/kg body weight per min for 15 min every other day with three times per week), and subsequent application of cyclic compressive loading (1200 cycles/day) on the tibiae of high-fat diet/streptozotocin-treated (HFD + STZ) mice. **b**, **c** Micro-CT data of cortical and trabecular bone architecture in proximal tibiae. **d** Three-point bending testing showing tibial whole-bone mechanical properties. **e**, **f** Immunohistochemical staining of the RANKL and OPG expression of osteocytes within tibial cortical bone matrix. **g** Toluidine blue staining to label osteoblasts on tibial trabecular bone surface. **h** The experimental protocol of the conditioned medium collection from high-glucose and high-fat (HGHF)-exposed MLO-Y4 osteocytic cells treated with ISTA (1 μM) in response to FSS stimulation (2 Pa), and its incubation in HGHF-exposed primary osteoblasts and RAW264.7 cells. **i** Western blotting assays showing the effects of the conditioned medium collected from HGHF-exposed MLO-Y4 cells treated with ISTA in response to FSS on the expression of osteogenic differentiation-related markers (including Col1a1, Osx, and Runx2) in HGHF-exposed primary osteoblasts. **j**, **k** The ALP activity assays, ALP staining, and Alizarin red staining in primary osteoblasts. **l** The gene expression of osteoclastogenesis-related markers, including cathepsin K, TRAP, NFATc1, and calcitonin receptor in HGHF-exposed RAW264.7 cells. **m** Western blotting assays of osteoclasts-related markers, including cathepsin K, TRAP, NFATc1, and calcitonin receptor. Graphs represent mean ± SD (**b**–**g**: $n = 8$ mice per group; **i**–**m**: $n = 6$ biologically independent replicates). **b**–**g** *$P < 0.05$, **$P < 0.01$ and ***$P < 0.001$ by two-way ANOVA with Bonferroni's post test. **j**–**l** *$P < 0.05$, **$P < 0.01$ and ***$P < 0.001$ by one-way ANOVA with Bonferroni's post test. Specific $P$ values are provided in the Source Data file. Scale bars: **e**–**g** 20 μm; **k** 50 μm.

combined with the pharmacological modulation of SERCA2, in protecting against fragility-associated fractures in patients with T2D.

## Methods

### Animals

Four-week-old male C57BL/6 J mice were purchased from the Animal Center of the Fourth Military Medical University, and were randomly assigned to the non-diabetes group and the experimentally-induced T2D group via the high-fat diet/streptozotocin treatment (HFD + STZ). Mice in the non-diabetes group were fed with a standard commercial chow diet (#DOSSYJY-001, DOSSY), and mice in the HFD + STZ group were fed with a high-fat diet with 60 kcal% fat (#D12492, Research Diets) throughout the experimental period. Mice at 8 weeks of age in the HFD + STZ group were fasted overnight and injected intraperitoneally once daily with 40 mg/kg STZ (Sigma-Aldrich) dissolved in citrate buffer (pH = 4.5) for 5 consecutive days[59]. In addition, eight-week-old male and female KK-Ay mice and the corresponding controls were purchased from Beijing Huafukang Bioscience Co. Ltd as a genetically spontaneous type 2 diabetic animal model, and fed with a high-fat diet throughout the experiment. Blood samples were collected from the tip of tail, and plasma glucose levels were determined with a glucometer (OneTouch SureStep Plus, Lifescan). Mice in T2D groups with the glucose concentration over 300 mg/dl were identified as the qualified animal models.

C57BL/6 mice with SERCA2 gene specifically overexpression in osteocytes (SERCA2$^{flox/flox}$; DMP1-Cre) were created based on the Cre/loxP system. DMP1-Cre transgenic mice (with the DMP1-Cre gene promoter drives Cre expression in osteocytes) were generated by GemPharmatech Co., Ltd. The floxed SERCA2 (SERCA2$^{flox/flox}$) mice were generated by Cyagen Biosciences. The SERCA2$^{flox/flox}$ mice were crossed with the DMP1-Cre mice to produce SERCA2$^{flox/+}$; DMP1-Cre mice. The offspring were then intercrossed to generate SERCA2$^{flox/flox}$; DMP1-Cre (termed *SERCA2* cKI) mice. Mice were fed with a high-fat diet, and subjected to daily STZ (40 mg/kg) intraperitoneal injection for 5 consecutive days since 8 weeks of age. Mice were housed under a 12-h light/dark cycle at an ambient temperature of 23 ± 1 °C and a relative humidity of 55 ± 5%. Animal studies were approved and carried out according to the guidelines of the Institutional Animal Care and Use Committee of the Fourth Military Medical University, and in compliance with ARRIVE guidelines.

### Tibial strain measurements

Strain levels on the antemedial surface of the tibia were measured in a separate cohort of mice in the non-diabetes, KK-Ay and HFD + STZ groups at 12 weeks of age ($n = 6$ tibiae/group). The musculature surrounding the tibiae was carefully removed, and a miniature single-element strain gauge (BX120-1AA, Yiyang) was attached to a relatively flat region on the antemedial surface of the tibia. A preload with 1 N was applied to immobilize the sample, and then the tibia was compressively loaded at 0.2 mm/s using a custom-designed mechanical loading device, which was described in our previous studies[36]. According to the recorded strain-force curves, the peak magnitude of cyclic compressive loading with 9 N for the non-diabetes group, 7.9 N for the HFD + STZ group, and 6.8 N for the KK-Ay group was applied to induce a consistent tensile strain (+1455 με) on the antemedial surface of the tibia.

### In vivo tibial mechanical loading

Mice at the age of 12 weeks in each group were anesthetized with 2% isoflurane, and then transferred to a custom-designed mechanical loading device (Fig. S2). The left tibiae were constrained and subjected to cyclic compression with a ramp loading waveform repeated at 4 Hz for 1200 cycles/day for a total of 2 weeks. The contralateral right tibia was not subjected to cyclic compressive loading and served as a control. To investigate the effects of SERCA2 on bone mechano-responsiveness in KK-Ay and HFD + STZ mice, the SERCA2 agonist ISTA was intravenously infused with 73 μg/kg body weight per min for 15 min every other day with three times per week from the first day of mechanical loading. Calcein (30 mg/kg; Sigma-Aldrich) and alizarin red (30 mg/kg, Sigma-Aldrich) were injected intraperitoneally at 10 days and 3 days before euthanasia by excessive inhalation of isoflurane, respectively.

### Micro-computed tomography scanning

After sacrifice, Bilateral tibiae were scanned using a micro-computerized tomography (micro-CT) machine (GE Healthcare) with an isotropic voxel resolution of 14 μm, an X-ray tube voltage of 80 kV, a tube electric current of 80 μA, and an exposure time of 2960 ms. Images were reconstructed and analyzed using the VGStudio Max software (Volume Graphics). The analyzed volume of trabecular bone was selected starting at 0.5 mm distal to the growth plate and extending 1 mm to the distal end (only contained the secondary spongiosa). The volume of interest for cortical bone analyses was selected starting at a distance 37% of the whole-bone length from its proximal end and extending 1 mm to the distal end. The following bone structural parameters were quantified, included trabecular bone volume fraction (BV/TV), bone surface to volume ratio (BS/BV), trabecular thickness (Tb.Th), trabecular number (Tb.N), trabecular separation (Tb.Sp), cortical thickness (Ct.Th) and cortical porosity (Ct.Po).

### Three-point bending tests

The three-point bending tests were performed on a universal mechanical testing machine (ElectroForce 3220, Bose Corp). The bone specimens were positioned horizontally on two supports with a span of 8 mm. The sample was immobilized with a preload of 1 N, and then deformed at a constant displacement rate of 10 mm/min until ultimate failure. The load-displacement curve was generated to quantify the parameters of whole-bone structural mechanical properties, including ultimate load (N), stiffness (N/mm), and energy to failure (mJ).

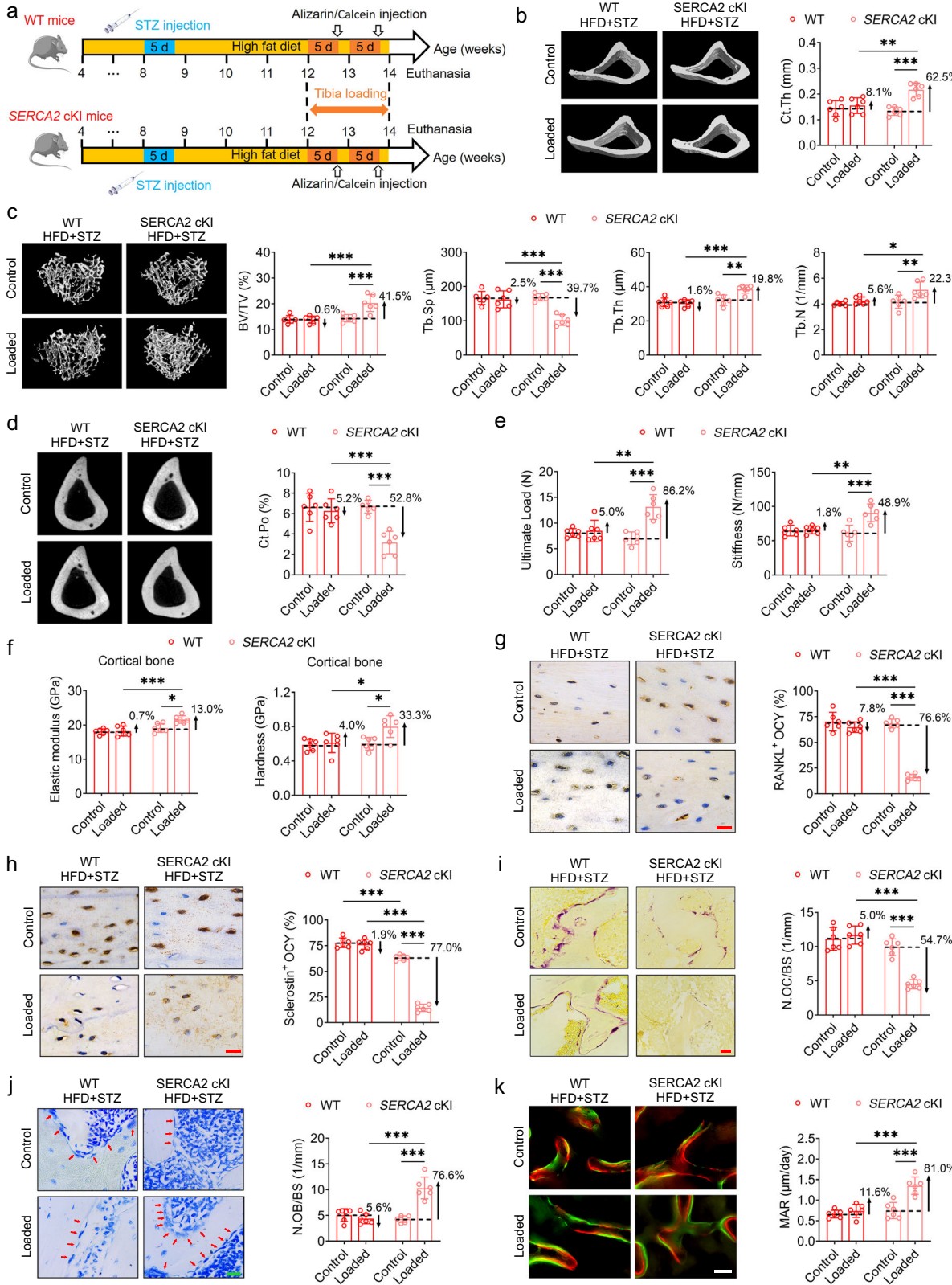

## Indentation testing

Samples were dehydrated in gradient ethanol, embedded in methyl methacrylate, and then transversely sectioned into a thickness of 1 mm. Then, specimens were polished sequentially using silicon carbide abrasive papers with the grit size of 800, 1000, and 1200 under water irrigation. After rehydration by immersing in saline solution for 24 h, samples were transferred to an indentation device (Agilent G200,

Agilent Technologies Inc.) with a Berkovich diamond tip. During testing, the tip was loaded into trabecular or cortical bone matrix to a final depth of 4 μm at a constant strain rate of 0.05 sec⁻¹, and then maintained for 10 sec in order to reduce the viscoelastic behavior and creep of bone specimens. Unloading was then applied to 10% of the peak load with the maximum loading rate, and maintained for 60 sec to determine the thermal drift. After removal of the tip from the

**Fig. 8 | Specific overexpression of SERCA2 in osteocytes suppresses the T2D-induced deterioration in bone mechano-responsiveness. a** The experimental protocol of the high-fat diet/streptozotocin treatment in SERCA2^flox/flox; DMP1-Cre (*SERCA2* cKI) mice, and subsequent application of cyclic compressive loading (1200 cycles/day) on the tibiae. **b–d** Micro-CT scanning for showing trabecular bone architecture, cortical bone thickness, and cortical porosity in the proximal tibiae of *SERCA2* cKI mice with high-fat diet/streptozotocin treatment subjected to subsequent exogenous cyclic loading. **e** Three-point bending testing showing whole-bone mechanical properties in the tibiae of *SERCA2* cKI mice with high-fat diet/streptozotocin treatment. **f** Nanoindentation testing showing local material properties in the tibiae of *SERCA2* cKI mice with high-fat diet/streptozotocin treatment.

**g, h** Immunohistochemical staining showing the protein expression of RANKL and sclerostin in osteocytes in tibial cortical bone matrix of *SERCA2* cKI mice with high-fat diet/streptozotocin treatment. **i** Representative TRAP staining to label osteoclasts on tibial bone surfaces, and the corresponding statistical result. **j** Representative toluidine blue staining to label osteoblasts on tibial trabecular bone surface, and the corresponding statistical result. **k** Representative calcein and alizarin red double labeling of tibial trabecular bone surfaces, and the corresponding statistical result. Graphs represent mean ± SD (*n* = 6 mice per group). *$P < 0.05$, **$P < 0.01$, and ***$P < 0.001$ by two-way ANOVA with Bonferroni's post test. Specific *P* values are provided in the Source Data file. Scale bars: **g**, **h**, **j** 20 μm; **i** 30 μm; **k** 50 μm.

specimen surface, the force-displacement curve was collected, and intrinsic material properties of bone, including the modulus and hardness, were calculated.

## Dynamic bone histomorphometry

Samples were dehydrated in gradient ethanol, embedded in methyl methacrylate, and then transversely cut into a thickness of 50 μm using a diamond saw in a microtome (Leica SpA). Calcein and alizarin red labeling were observed on both trabecular and cortical bone surfaces in similar regions as micro-CT analyses under a fluorescence microscope (IX83, Olympus). Histomorphometric parameters were measured using the ImageJ software (National Institutes of Health), including single-label perimeter (sL.Pm), double-labeled perimeter (dL.Pm), bone perimeter (B.Pm), and interlabel width (Ir.L.Wi). Based on the above primary data, the following parameters were calculated, including mineral apposition rate (MAR = Ir.L.Wi/7 days), mineralizing surface per bone surface (MS/BS = [0.5 × sL.Pm+dL.Pm]/B.Pm × 100), and bone formation rate per bone surface (BFR/BS = MAR × MS/BS).

## Bone histological analyses

After dissection, tibial samples were fixed in 4% paraformaldehyde (PFA), demineralized in 10% ethylenediaminetetraacetic acid (EDTA), dehydrated in gradient ethanol, embedded in paraffin, and then cut longitudinally into 5-μm-thick sections. Sections were stained with tartrate-resistant acid phosphatase (TRAP, #387-1KT, Sigma-Aldrich) to quantify the number of osteoclast per millimeter of bone surface (N.OC/BS) according to the manufacturers' protocols. Samples were stained with toluidine blue (#DA0059, Leagene Biotechnology) to calculate the number of osteoblast per millimeter of bone surface (N.OB/BS), and stained with hematoxylin and eosin (H&E; Servicebio) to calculate the number of bone marrow adipocytes and the number of the empty osteocytic lacunae in bone matrix. Ploton silver staining was performed to visualize the morphology of osteocyte canalicular network, and the canalicular density and length in tibial bone matrix were quantified.

## Immunohistochemistry

The embedded sections were deparaffinized in xylene, rehydrated with gradient ethanol, pre-treated in a microwave with 10 mM citric acid buffer (pH = 6.0) for 15 min for antigen retrieval, and then treated with a 3% hydrogen peroxide solution for 25 min to inhibit endogenous peroxidase activity. Then, sections were blocked with 3% bovine serum albumin (BSA) for 30 min, and incubated with the anti-Caspase-3 antibody (1:200; Cell Signaling Technology), anti-RANKL antibody (1:100; Proteintech), anti-OPG antibody (1:100; Abcam), anti-Sclerostin antibody (1:200; R&D systems), anti-SERCA1 antibody (1:100; Affinity Biosciences), rabbit anti-SERCA2 antibody (1:100; Abcam), rabbit anti-SERCA3 antibody (1:100; Proteintech) or rabbit anti-PPARα (1:200; Affinity Biosciences) at 4 °C. Sections were then incubated with the corresponding horseradish peroxidase-labeled secondary antibody, and detected using the 3,3'-diaminobenzidine (DAB) kit (Servicebio). Sections were counterstained with hematoxylin and photographed under the Olympus optical microscope.

## Serum biochemical analysis

Serum samples were obtained from anesthetized mice by cardiac puncture, and then stored at −80 °C until use. According to the manufacturer's instructions, serum levels of P1NP, CTX-1 and sclerostin were measured using the commercially available enzyme-linked immunosorbent assay (ELISA) kits.

## Osteocyte Ca²⁺ imaging in situ in the tibia under cyclic loading

Twelve-week-old C57BL/6 J mice were euthanized by excessive inhalation of isoflurane. Bilateral intact tibiae were quickly extracted under sterile conditions, and incubated in phenol red-free α-MEM (Gibco) supplemented with 5% fetal bovine serum (FBS, Gibco), 5% calf serum (CS, Gibco), and 1% penicillin/streptomycin (P/S, Gibco) at 37 °C for 1 h. Tibiae were immersed in phenol red-free α-MEM supplemented with 4.6 μM Ca²⁺ indicator Calbryte-520 AM (AAT Bioquest) dissolved in DMSO, 0.04% pluronic acid F-127, and 2 mM probenecid for 1 h, and then transferred to a custom-designed mechanical loading device specially used for cellular Ca²⁺ signaling studies in ex vivo bone tissues (Fig. S2). The anteromedial surface of the tibia was observed under a confocal microscope (Fluoview FV3000, Olympus) with a 10× objective at 488 nm laser excitation. A unique synchronized cyclic loading/confocal imaging technique was employed to avoid the shift of the confocal focus during cyclic loading. A total of 115 cycles of axial compressive loading was applied on the tibiae in the mice of non-diabetes, HFD + STZ, and KK-Ay groups with magnitude of 9 N, 7.9 N, and 6.8 N, respectively, which corresponded to identical 1455 με tensile strain on the antemedial surface (the Ca²⁺ imaging region). For image acquisition, a dwell time of 3.9 s was applied between each loading cycle. Confocal images were captured at 1.1 s per frame for 120 frames containing 5 frames of baseline control prior to mechanical loading. The parameters associated with Ca²⁺ dynamics were determined using the Fluoview FV31S-SW software (Olympus), including the percentage of responsive cells, the number of Ca²⁺ spikes, average Ca²⁺ intensity, and Ca²⁺ spike relaxation time.

## Cell culture and high glucose/high fat treatment

The murine osteocyte-like MLO-Y4 cell line was kindly provided by Dr. Lynda Bonewald (University of Missouri). MLO-Y4 cells were seeded into cell culture dishes coated with rat tail type I collagen (Corning), and grew in α-MEM (Gibco) supplemented with 5% FBS, 5% CS, and 1% P/S at 37 °C. Primary osteoblasts were isolated from the mouse calvaria by triple collagenase/Dispase II digestion. Cells were cultured in α-MEM containing 10% FBS and 1% P/S. For the induction of osteoblast differentiation, cells were cultured in α-MEM containing 10% FBS and osteogenic induction supplements (50 μg/ml ascorbic acid and 4 mM β-glycerophosphate). The RAW264.7 cells, purchased from ATCC, were cultured in DMEM (Gibco) supplemented with high glucose (4.5 g/L) and 10% FBS at 37 °C. The RAW264.7 cells were induced to differentiate into osteoclast-like cells by treating with 50 ng/ml RANKL (R&D systems). Cells in the high glucose and high-fat (HGHF) group was incubated in the cell culture medium with 25 mM glucose (Sigma-Aldrich) and 0.2 mM palmitic acid (Sigma-Aldrich). Cells in the mannitol-treated group (MAN) were cultured in the cell culture

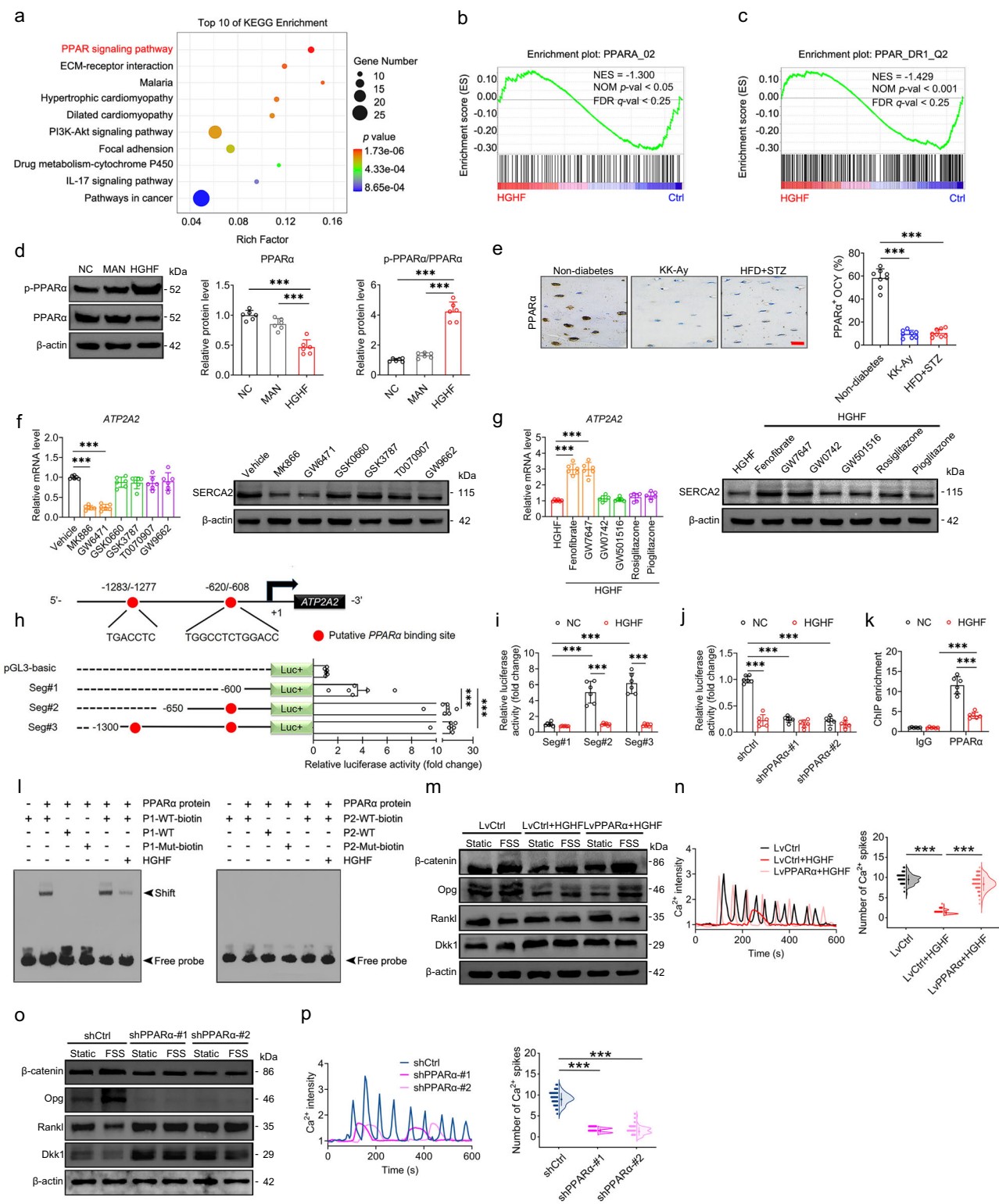

medium with 19.7 mM mannitol as the an osmolarity control, and cells in the control group were incubated in the normal medium with 5.5 mM glucose.

## RNA silencing and overexpression

For the knockdown of *SERCA2*, the MLO-Y4 osteocytic cells were transfected with siRNA targeting mouse *SERCA2* (GenePharma Co., Ltd) or control siRNA using the GP-siRNA-Mate Plus reagent (Gene-Pharma Co., Ltd) according to the manufacturer's protocol. The

knockdown efficacy was confirmed using western blotting at 48 h after transfection. To construct the stable PPARα knockdown cell line, the LV2 (pGLVU6/Puro) lentiviral vector was purchased from GenePharma Co., Ltd, and transfected into MLO-Y4 cells to generate PPARα-specific shRNA or shCtrl. For the establishment of the cell line that stably overexpressed PPARα, the LV6 (EF-1a/Puro) lentiviral vector (Gene-Pharma Co., Ltd) containing PPARα was used to transfect cells. Cells transfected with lentiviral particles carrying the empty vector served as the negative control. Fresh medium containing puromycin (8 μg/mL)

**Fig. 9 | The nuclear transcription factor PPARα mediates T2D-induced specific reduction in the expression of osteocytic SERCA2 pump. a** RNA-seq-based KEGG pathway analysis showing the top 10 enriched KEGG pathways. **b, c** GSEA analysis showing a significant enrichment of signaling events associated with PPARα and PPARDR1_Q2. **d** The PPARα and p-PPARα protein expression in osteocytes.
**e** Immunohistochemical staining of osteocytic PPARα in diabetic and non-diabetic tibiae. **f** The SERCA2 expression in osteocytes treated with antagonists of PPARα (MK886 and GW6471), PPARβ/δ (GSK0660 and GSK3787), and PPARγ (T0070907 and GW9662). **g** The SERCA2 expression in HGHF-exposed osteocytes treated with agonists of PPARα (Fenofibrate and GW7647), PPARβ/δ (GW0742 and GW501516), and PPARγ (rosiglitazone and pioglitazone). **h** A schematic representation and the relative luciferase activities of three *ATP2A2* promotor regions. **i** The relative luciferase activity assays of Seg#1, Seg#2 and Seg#3. **j** The relative luciferase activity in HGHF-treated osteocytes with PPARα silencing. **k** ChIP assays showing the PPARα enrichment on *ATP2A2* promotor in normal and HGHF-treated osteocytes. **l** EMSA

assays confirming the binding of PPARα to the *ATP2A2* promoter region (−620 to −608 bp). The nuclear extract were incubated with biotin-labeled wild-type (WT-biotin) probe, unlabeled wild-type (WT) probe, and biotin-labeled mutated (Mut-biotin) probe. Red letters indicate substituted nucleotide sequences in the mutated probes (P1: −620 to −608 bp; P2: −1283 to −1277 bp). **m, n** Intracellular Ca$^{2+}$ signaling and protein expression of osteocyte-related cytokines in HGHF-treated osteocytes with PPARα overexpression subjected to FSS. **o, p** Intracellular Ca$^{2+}$ signaling and the expression of osteocyte-related cytokines in MLO-Y4 cells with lentiviral silencing of PPARα subjected to FSS. Graphs represent mean ± SD (**d, f, g, i–k, m, o** $n = 6$ biologically independent replicates; **e** $n = 8$ mice per group; **l** $n = 3$ independent replicates; **n, p:** $n = 120$ cells per group). **a** $P$ value was obtained by one-tailed hypergeometric test. **b, c** $P$ values were obtained by one-tailed permutation test. **d–h, n, p** ***$P < 0.001$ by one-way ANOVA with Bonferroni's post test. **i–k** ***$P < 0.001$ by two-way ANOVA with Bonferroni's post test. Specific $P$ values are provided in the Source Data file. Scale bars: **e** 20 μm.

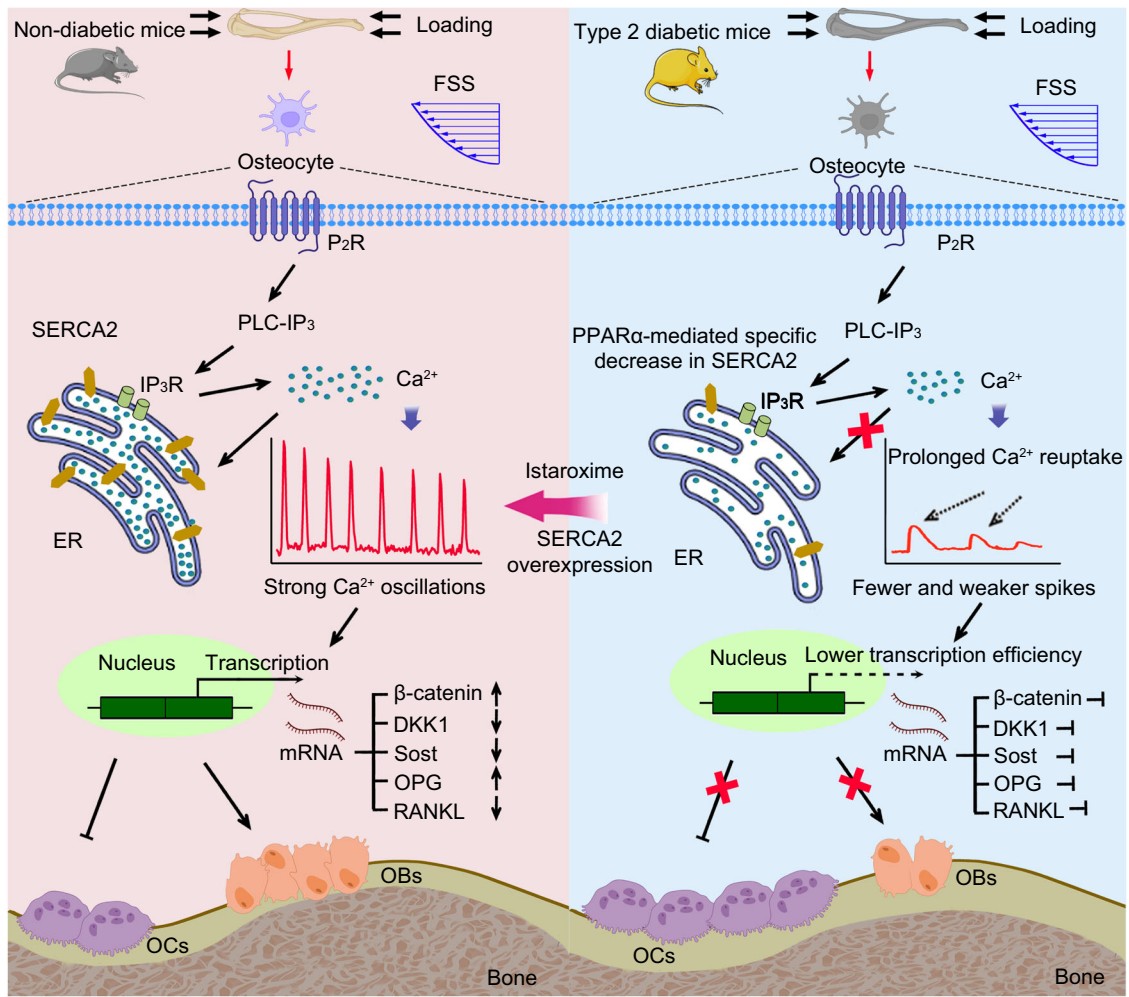

**Fig. 10 |** Schematic drawing of the mechanisms whereby T2D weakens bone mechano-responsiveness, and rescuing SERCA2 pump deficiency enhances exogenous cyclic loading-mediated bone gain in T2D.

was added to select for stably infected cells. The silencing or over-expression efficiency was assessed via western blotting.

## Osteocyte/osteoblast Ca$^{2+}$ imaging in vitro under fluid flow stimulation

MLO-Y4 cells or primary osteoblasts were seeded on rat tail type I collagen-coated glass slides and cultured at 37 °C overnight. Then, cells were incubated with 2.3 μM Ca$^{2+}$ indicator Calbryte-520 AM at 37 °C for 1 h. The glass slide with the cell side facing downward was mounted into a parallel-plate flow chamber, and observed under the

confocal microscope with a ×20 objective at 488 nm laser excitation. Cells were exposed to laminar steady fluid flow using the peristaltic pump or oscillatory fluid flow at 1 Hz driven by a Hamilton glass syringe[60]. Confocal images were collected and the intracellular Ca$^{2+}$ signaling was quantitatively analyzed using the Fluoview FV31S-SW software.

## The apoptosis and viability assays of osteocytes

The apoptosis assays were performed using flow cytometry with an Annexin V-FITC/PI apoptosis detection kit (KeyGEN). After

centrifugation, cells were resuspended in 500 μl binding buffer containing 5 μl Annexin V-FITC and 5 μl propidium iodide (PI), and incubated in the dark for 15 min. The Beckman-Coulter XL flow cytometer was used for the measurements with the emission at 530/630 nm. For the cell viability assays, the Cell Counting Kit-8 (CCK-8; #C0037, Beyotime) was added into the MLO-Y4 cells, and then incubated at 37 °C for 2 h. The absorbance was determined using a microplate reader (Bio-Rad) at 450 nm.

## Alkaline phosphatase (ALP) staining and activity assays of osteoblasts

Primary osteoblasts were fixed with 4% PFA for 15 min, and then incubated with substrate solution for the reaction of ALP at 37 °C for 1 h using a commercial ALP staining kit according to the manufacturer's instructions (Beyotime). The ALP activity was assessed with a commercial ALP kit (Nanjing Jiancheng Bioengineering Institute) on the total proteins extracted using the radioimmunoprecipitation assay (RIPA) buffer.

## Alizarin red staining of osteoblasts

Primary osteoblasts were stained with 2% Alizarin Red S solution (Beyotime) for 30 min. The stained images were captured using the microscope. For the quantitative assessment of the degree of mineralization, the red stain was eluted by 10% acetic acid for 30 min, and then quantified via spectrophotometric absorbance measurements of the optical density at 590 nm.

## RNA-seq and bioinformatics

Total RNA from the MLO-Y4 cells treated with normal glucose or HGHF was isolated using the TRIzol reagent (Sigma-Aldrich). The RNA integrity was determined using the Bioanalyzer 2100 (Agilent Technologies Inc.), and the concentration and purity of RNA were measured using the NanoDrop ND-1000. After quality control and purification, the transcripts were reverse-transcribed into cDNA by SuperScript™ II Reverse Transcriptase to create the cDNA library. RNA sequencing was performed on the NovaSeq 6000 system (LC-Bio Technology CO., Ltd.). The gene expression was analyzed using the FKPM method by StringTie and ballgown (http://www.bioconductor.org/packages/release/bioc/html/ballgown.html). R package edgeR or DESeq2 algorithm was used to select differentially expressed genes (with fold change>2 or <0.5 and $P < 0.05$). Gene ontology (GO) enrichment analysis, Kyoto Encyclopedia of Genes and Genomes (KEGG) enrichment analysis, and the gene-set enrichment analysis (GSEA) were carried out to identify significantly up- and down-regulated genes and related pathways those were highly enriched.

## Quantitative real-time PCR (qRT-PCR)

Samples (tibial tissues or cells) were homogenized in TRIzol reagent (Sigma-Aldrich). Total RNA was isolated using the guanidinium isothiocyanate-alcohol phenyl-chloroform method following the manufacturer's protocol, and then quantified using Denovix DS-11 spectrophotometer (Denovix). The RNA samples were reverse-transcribed to cDNA using the cDNA synthesis kit (#RR036A, Takara) according to the manufacturer's instructions. cDNA was amplified with TB Green Premix Ex Taq II (Takara) using the LightCycler 480 Real-time PCR System (Roche Applied Science). The primer pairs are shown in Table S1. The levels of mRNA were normalized to the β-actin housekeeping gene, and the gene expression was calculated according to the $2^{-\Delta\Delta Ct}$ method.

## Western blotting

Total cellular proteins were extracted using the RIPA lysis buffer, and a BCA assay kit (Beyotime) was used to quantify the protein concentrations. The same amounts of proteins (20 μg) were separated by polyacrylamide gel electrophoresis (SDS-PAGE) and transferred onto PVDF membranes (Millipore). Membranes were blocked in 5% low-fat milk for 1 h, and then incubated with primary antibodies against SERCA1 (1:500; Affinity Biosciences), SERCA2 (1:5000; Abcam), SERCA3 (1:5000; Proteintech), IP3R (1:1000; Affinity Biosciences), PLCβ1 (1:1000; Proteintech), P2Y2 (1:1000; Affinity Biosciences), P2Y4 (1:1000; Affinity Biosciences) P2Y12 (1:2000; Abcam), Col1a1 (1:1000; Abcam), Runx2 (1:1000; Affinity Biosciences), Osx (1:1000; Bioss), β-catenin (1:1000; Proteintech), OPG (1:1000; Abcam), RANKL (1:1000; Proteintech), DKK1 (1:2000; Proteintech), NFATc1 (1:500; Santa Cruz), Calcr (1:1000; Proteintech), Ctsk (1:1000; Affinity Biosciences), TRAP (1:2000; Abcam), PPARα (1:1000; Affinity Biosciences), p-PPARα (1:1000; Affinity Biosciences) or β-actin (1:4000; Proteintech) overnight at 4 °C. The blots were incubated with HRP-conjugated goat-anti-rabbit secondary antibody (1:5000; Abcam) or HRP-conjugated goat-anti-mouse secondary antibody (1:5000; Abcam), and then visualized using an enhanced chemiluminescence system (Image Quant 350, GE Healthcare).

## The mRNA and protein stability assays

For the mRNA stability assays, MLO-Y4 cells exposed to normal glucose or HGHF were treated with 10 μg/ml actinomycin-D (MedChemExpress) for 0, 2, 4, and 6 h, respectively. Total RNA was isolated from samples at various time points and analyzed with qRT-PCR to determine the *ATP2A2* mRNA levels. For the protein stability assays, MLO-Y4 cells after treatment with normal glucose or HGHF were incubated with 50 μg/ml cycloheximide (MedChemExpress) for 0, 2, 4 and 6 h, respectively. Total protein was extracted, and the expression of SERCA2 was then analyzed by western blotting.

## Transcription factor prediction and luciferase reporter gene assays

The genomic sequence of *ATP2A2* with 3000 bps was downloaded from the website of http://asia.ensembl.org/index.html. The prediction for the possible binding sites of the downloaded top 3000 bases and transcription factors screened by the GSEA analysis were performed on http://bioinfo.life.hust.edu.cn/AnimalTFDB/#!/tfbs_predict, and the base sequences with the P value lower than $10^{-5}$ were obtained. By integrating the GSEA analysis with transcription factor prediction, the transcription factors that might recognize the transcription initiation site of *ATP2A2* were identified. The identified *ATP2A2* transcription initiation site was truncated at the binding sites of −1300, −650 and −600, and then transfected into cells with pGL3-basic vector. To identify the specific transcription factor sequence binding to *ATP2A2* promoter in the predicted sequences, the relative luciferase activity of the three truncated sequences of the transcription factor PPARα were determined using a luciferase assay kit (Beyotime). The luciferase activity was normalized against Renilla luciferase value. Plasmid PGL3-basic served as a negative control.

## ChIP assays

ChIP assays were conducted using an EZChIP™ kit (Millipore) following the procedures provided by the manufacturer. The MLO-Y4 cells were incubated with 1% formaldehyde for 10 min at room temperature to cross-link DNA and then sonicated in lysis buffer. The immunoprecipitation was performed using the antibody against PPARα (Affinity Biosciences) or IgG. After washing, the complexes were eluted with 100 μl of chelating resin solution and boiled for 10 min. The purified DNA was subjected to PCR targeting the *ATP2A2* promotor with primers (sequences are shown in Table S1). Precipitated genomic DNA was amplified by qRT-PCR using SYBR Green PCR Master Mix.

## EMSA assays

EMSA was performed to assess the transcription factor PPARα and *ATP2A2* DNA binding activity using the LightShift® Chemiluminescent EMSA Kit (Thermo Fisher Scientific). Biotin-labeled probes were

incubated with nuclear extracts in a 20 μl of the EMSA reaction buffer containing 2 μl of 10× binding buffer, 1 μg of poly (dI-dC), 50% glycerol, 1% NP-40, and 100 mM $MgCl_2$ for 20 min. For competitive binding assays, excessive biotin-unlabeled probes were added to the EMSA reaction system under the same condition. The binding complexes were electrophoresed on 6.5% polyacrylamide gel at 150 V for 45 min. The probe sequences used are listed in Table S2.

## Statistical analysis

All data are shown as the mean ± S.D. Statistical analysis was performed using the Prism version 9.5.1 for Windows (GraphPad). Two-tailed Student's *t* test was used to compare the difference between two groups. One-way analysis of variance (ANOVA) was performed for comparisons between more than two groups for one condition. Two-way ANOVA was used to determine significant differences for two conditions. Bonferroni's post hoc test was employed following ANOVA. Differences were considered significant at $*P < 0.05$, $**P < 0.01$, and $***P < 0.001$.

## Reporting summary

Further information on research design is available in the Nature Portfolio Reporting Summary linked to this article.

## Data availability

The RNA-seq data have been deposited to SRA (Sequence read archive) under the accession number PRJNA901004 that is publicly accessible at https://www.ncbi.nlm.nih.gov/search/all/?term=PRJNA901004. The genomic sequence of ATP2A2 was obtained from the Ensembl Genomes databases (http://asia.ensembl.org/index.html). The remaining data are available within the paper and Supplementary Information. Source data are provided with this paper.

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

## Acknowledgements

The authors acknowledge the support from The National Natural Science Foundation of China (11972366 and 12172378 to D.J., 51907111 to J.C., 51777211 to E.P.L., and 12302412 to Z.D.Y.), the Young Talent Supporting Project in Military Technology of China (17-JCJQ-QT-038 to D.J.), and the Shaanxi Provincial Natural Science Foundation of China (2021SF-023 to J.C.). The authors thank Bin Wang and Xingfan Wu (College of Basic Medicine, Chongqing Medical University of China) for their help in replicating the data of mechanical loading-mediated changes of canalicular length and density in normal mice. For the creation of images, authors also extend their thanks to BioRender.com (https://biorender.com/) and Servier Medical Art provided by Servier (https://smart.servier.com/), licensed under a Creative Commons Attribution 3.0 unported license.

## Author contributions

D.J., X.E.G., J.C., L.L.S., W.J.L., and E.P.L. designed research. X.S., Y.L.T., J.L., P.L., Z.D.Y., X.X.H., D.W., and Y.J.D. performed research. X.S., Y.L.T., J.L., Z.D.Y., P.L., J.C., L.L.S., X.X.H., and D..J. analyzed data. X.S. and D.J. wrote the paper.

## Competing interests

All the authors declare no competing interests.
