## [Peer Review File · Nature Communications]

Rescuing SERCA2 pump deficiency improves bone mechanoresponsiveness in type 2 diabetes by shaping osteocyte calcium dynamicsREVIEWER COMMENTS

Reviewer #1 (Remarks to the Author):

This is a potentially interesting study revealing a molecular mechanism regulating bone mechanosensitivity in T2D. The authors provide strong evidence for a role of SERCA2 in osteocytes, which may mediate T2D-induced deterioration of bone mechanosensitivity. However, the evidence for involvement of PPAR α for T2D-induced downregulation of SERCA2 in osteocytes is less convincing.

Major points:

1. The authors showed that HGHF decreased SERCA2 at both mRNA and protein levels in osteocytic MLO-Y4 cells (Figure 4c-d). Moreover, they showed that SERCA2 expression was decreased at both mRNA and protein levels in diabetic mouse models (Figure 4e-f). The authors ascribed such changes to altered Atp2a2 transcription in osteocytes. However, it is possible that Atp2a2 mRNA stability was decreased in osteocytes in T2D, which might contribute to the diminution of Atp2a2 mRNA level. Moreover, in both cases, SERCA2 protein exhibited a more pronounced decrease as compared to its mRNA, which suggests that SERCA2 protein might become less stable under such conditions. The authors need to investigate whether the stability of SERCA2 is altered at mRNA and protein levels in osteocytes under diabetogenic conditions.
2. The data on PPAR α in regulation of SERCA2 is confusing. The authors show that the PPAR α agonists increased, while the antagonists decreased, the protein levels of SERCA2 in osteocytes (Figure 9d-e). They further demonstrate that PPAR α binds to Atp2a2 promoter in a CHIP assay, which was enhanced under HGHF condition (Figure 9i). In a luciferase assay, Atp2a2 promoter containing putative PPAR α binding sites was activated under HGHF condition, which is indeed dependent on PPAR α (Figure 9f-h). Based on the data presented in Figure 9f-i, one would predict that Atp2a2 transcription would be enhanced in osteocytes under HGHF condition, which would be the opposite to what the authors observed in Figure 4c-d. Again, these data might suggest that Atp2a2 mRNA stability might be affected.
3. The authors should provide experimental evidence for PPAR α expression in osteocytes under diabetogenic conditions *in vitro* and *in vivo*, to see whether its mRNA or protein is decreased.
4. In Fig S6a-b, only one siRNA for Atp2a2 and one shRNA for PPAR α caused a significant reduction of the target proteins. In Fig S6c, only one PPAR α overexpression clone exhibited a significant increase of PPAR α expression. The authors need to repeat the functional assays with an independent siRNA/shRNA or an independent PPAR α overexpression clone.
5. The FSS/loading-induced calcium oscillation in osteocytes is maintained by both SERCA2 and IP3R as shown by the authors. Unfortunately, the potential involvement of IP3R in T2D-induced impairment of calcium oscillation in osteocytes and deterioration of bone mechanosensitivity has not been properly addressed. Please clarify whether the P2R-PLC-IP3-IP3R pathway is altered in osteocytes in diabetes or not.

Major points:

1. The western blots should be quantified across the MS.
2. The experiments were only performed on male mice. It would be nice to show whether this mechanism is also critical in female mice.
3. The putative PPAR α binding sites were found in Atp2a2 promoter. The authors need to demonstrate whether PPAR α indeed binds to these sites via a gel shift experiment.
4. In Fig 8g-i, the authors indicate n=6. Please clarify whether this means images from 6 mice or 6 images from one mouse. They need to clarify such information in many other figures throughout the MS.
5. On Page 19, the author wrote that 'we found that the T2D-induced reduction of bone mechanosensitivity was attributed to the increased phosphorylation of PPAR α -mediated specific reduction in the expression of osteocytic SERCA2'. They need to present the data on phosphorylation of PPAR α .

Reviewer #2 (Remarks to the Author):

Shao et al., describe the role of SERCA2 in the lack of anabolic response of osteocytes to error loading. It is an elegant study, combining validated in vivo models with well-established cell culture experiments in order to distinguish between systemic effects of T2D on bone vs. direct effects on osteocytes. En passant the study clearly demonstrates the importance of including mechanical stimuli in bone research (e.g. fig 6 ISTA does not affect bone parameters in the absence of mechanical stimulation). Overall a substantial amount of work has been performed, leading to an accumulation of convincing results. The results are interesting for a large audience, and are likely to lead to societal impact.

I did not find a specifically formulated goal or hypothesis. This is unfortunate, because I wonder whether the proper loading regime was applied.

The tibia loading model creates a positive error load in one tibia, that exceeds the daily mechanical loads generated by the physical activity of, in this case, mice going about their business. The control tibia is thus not "unloaded", but rather "normally loaded". An error load creates a gain in bone mass in the loaded bone, as nicely shown in several figures in the current manuscript. The error load can be compared to a human whom will take up jogging (at an easy pace), but may not represent normal activity. Thus, the lack of anabolic response in loaded tibia in T2D mouse models indicate that exercise will not benefit bone properties in people with T2D, but it does not necessarily explain the occurrence of the bone phenotype in T2D.

Istaroxime was effective in loaded bones (people who take up jogging), but did not affect the control bones (people who go about their normal daily activities). Thus, following the paper, potential therapeutic effects should be based on a combination of an exercise regime and taking SERCA2 agonists.

Experiments in which hindlimbs of T2D mice are unloaded and bone parameters compared to those of normal ambulatory mice would generate significant additional insights. Without such experiments the introduction needs some rewriting to avoid the suggestion that the etiology of bone fragility in normal ambulatory T2D patients is investigated.

The first paragraph of the introduction is suggestive that people with T2D have less bone, similar to people who experience unloading for prolonged periods of time. However, most studies do not find a reduction in BMD in T2D in humans. On the contrary, increases in BMD in T2D have been regularly reported, although cortical porosity also clearly increases (as described in ref 5 of the manuscript). A recent meta analysis (Walle et al., current osteoporosis rep 2022) even goes so far as to claim that T2DM effects were more severe at the radius than the load-bearing tibia, supporting mechanoregulation of bone fragility in diabetes mellitus.

Also, unloading is associated with an increased turnover with a negative balance resulting in bone loss (as observed in the current study in fig 2 for the T2D models) while T2D in humans is generally associated with decreased turnover, resulting in a phenotype mimicking advanced skeletal ageing. I missed the link with the ageing phenotype and increased senescence in the current paper.

The authors regularly mention “mechanosensitivity”, suggesting that the setpoint of the mechanostat in osteocytes is changed with T2D, whereby osteocytes need a different intensity stimulus to respond. This is not what has been investigated. The magnitude of the response to 1 type of load has been investigated, which corresponds to a change in “mechanoresponsiveness”. In the abstract both words are used.

The materials and methods seem sound. The number of independent experiments should be mentioned. Why did the authors use a diamond saw to make sections rather than a microtome?

Sentence 70 is difficult to follow.

Sentence 55: Vertebra are not part of the lower extremities.

Reviewer #3 (Remarks to the Author):

Manuscript Number: NCOMMS-22-46839

The current manuscript submitted by Shao and colleagues, shows here that Calcium dynamics is important for the mechanosensitive function of osteocytes during mechanical loading. Mechanical loading failed to improve osteocyte function or bone parameters in models of Type 2 diabetes due to lack of response to calcium signaling as a result of downregulation of SERCA2. Serca2 antagonists suppressed

Calcium signaling in osteocytes, while Serca2 agonists, decrease cortical porosity, improved strength, and improved net bone formation. The study is well designed, and data mostly supports the hypothesis that SERCA2 is crucial for osteocyte mechanosensitivity to loading. However, there are certain deficiencies that are pointed out here.

1. Please quantify adipocyte numbers and volume per bone volume since Figure 1e suggest there are changes in adipocytes.
2. The data in Figure 2a, showing empty lacuna in KK-Ay and HFD-STZ mice does not match with staining patterns in 2b, 2c, 2d and 2e where the empty lacuna is not seen. How is it possible that one sample from an animal has empty lacuna, while others do not. Furthermore, these empty lacunae are not seen in Figures 4f, 6f, 6g, 6h and 6i. This is not consistent with what the authors have shown in 2a. Please explain these discrepancies.
3. What about osteocyte empty lacunae data following ISTA treatment in KK-Ay mice? That should be presented.
4. Osteocyte canalicular network is a key measure of osteocyte health and has direct indication for mechnanosensation. Hence this makes canalicular networks as a key piece of data which should be shown in both models of Typ2 diabetes and following SERCA2 agonist treatment.
5. Dynamic histomorphometry was shown in the paper, however static histomorphometry of osteoblast numbers was missing. That should be shown as well. This will support the data from dynamic histomorphometry.
6. One key data missing in all experimental design was markers for bone formation (P1NP), resorption (CTX-I) and sclerostin in the serum. This should be presented.

Response to Reviewers

Manuscript No. NCOMMS-22-46839A

Below are our responses (in BOLD type) to the Referee's comments on a point-by-point basis.

Comments from Reviewer(s):

Reviewer #1

This is a potentially interesting study revealing a molecular mechanism regulating bone mechanosensitivity in T2D. The authors provide strong evidence for a role of SERCA2 in osteocytes, which may mediate T2D-induced deterioration of bone mechanosensitivity. However, the evidence for involvement of PPARa for T2D-induced downregulation of SERCA2 in osteocytes is less convincing.

Major points:

1. The authors showed that HGHF decreased SERCA2 at both mRNA and protein levels in osteocytic MLO-Y4 cells (Figure 4c-d). Moreover, they showed that SERCA2 expression was decreased at both mRNA and protein levels in diabetic mouse models (Figure 4e-f). The authors ascribed such changes to altered Atp2a2 transcription in osteocytes. However, it is possible that Atp2a2 mRNA stability was decreased in osteocytes in T2D, which might contribute to the diminution of Atp2a2 mRNA level. Moreover, in both cases, SERCA2 protein exhibited a more pronounced decrease as compared to its mRNA, which suggests that SERCA2 protein might become less stable under such conditions. The authors need to investigate whether the stability of SERCA2 is altered at mRNA and protein levels in osteocytes under diabetogenic conditions.

Response: Thanks so much for this insightful and excellent comment and suggestion. We do agree with the reviewer's point that the mRNA and protein stability assays are really necessary for our current study. Thus, we have supplemented the relevant experiments during the manuscript revision period according to the reviewer's suggestion, and our new results reveal that the stability of SERCA2 was not significantly altered at either the mRNA or protein level in HGHF-exposed osteocytes. These new data and the corresponding description have been added into the revised manuscript (see Figure S6, Results L227-229, Methods L786-L793, and Supplemental Information L87-L90). It is really a quite valuable suggestion, which is helpful for making the experimental evidence of our manuscript more solid and convincing.

2. The data on PPAR α in regulation of SERCA2 is confusing. The authors show that the PPAR α agonists increased, while the antagonists decreased, the protein levels of SERCA2 in osteocytes (Figure 9d-e). They further demonstrate that PPAR α binds to Atp2a2 promoter in a ChIP assay, which was enhanced under HGHF condition (Figure 9i). In a luciferase assay, Atp2a2 promoter containing putative PPAR α binding sites was activated under HGHF condition, which is indeed dependent on PPAR α (Figure 9f-h). Based on the data presented in Figure 9f-i, one would predict that Atp2a2 transcription would be enhanced in osteocytes under HGHF condition, which would be the opposite to what the authors observed in Figure 4c-d. Again, these data might suggest that Atp2a2 mRNA stability might be affected.

Response: Many thanks for this insightful and professional comment. Our group has carefully analyzed Figure 9 as shown in our original manuscript, and we have also carefully doublechecked the relevant original experimental data. We found that our graduate student had mistakenly labelled the data of luciferase reporter gene assays and ChIP experiments during the process of graph preparation and plotting (That is, Figure 9g-i in the original manuscript provided an opposite labelling between the NC group and HGHF group). In fact, our results reveal that the luciferase activity was found to be significantly decreased in HGHF-exposed MLO-Y4 cells, and suppressing the PPAR α expression resulted in no significant difference in the luciferase activity between the NC group and the HGHF group. The ChIP assays reveal that the enrichment of PPAR α on ATP2A2 promoter was significantly decreased in HGHF-treated MLO-Y4 cells. The corresponding images have been corrected in the revised manuscript (see Figure 9i-k). Moreover, our newly added results of electrophoretic mobility shift assays (EMSA) provide further evidence to confirm that binding of PPAR α to the Atp2a2 promoter was significantly decreased under the HGHF condition (see Figure 9l). We do apologize that our original manuscript provided quite misleading information, and thanks so much for this valuable comment from the reviewer, which is really helpful for greatly improving the quality of our manuscript.

3. The authors should provide experimental evidence for PPAR α expression in osteocytes under diabetogenic conditions in vitro and in vivo, to see whether its mRNA or protein is decreased.

Response: Thanks so much for this professional and insightful suggestion. We do agree with the reviewer's point that the experimental

evidence for the PPAR α expression in osteocytes under diabetogenic conditions are essential for our current study. Thus, we have supplemented extra experiments to examine the protein expression of PPAR α in HGHF-exposed osteocytes *in vitro*, and also to assess the PPAR α protein expression in osteocytes of T2D skeletons *in vivo*. These new data have been added into the revised manuscript (see Figure 9). The corresponding results and methodology descriptions have been added into the revised manuscript (see Results L394-L398, Methods L655, L782-L783 and Figure Legends L1178-L1180).

4. In Fig S6a-b, only one siRNA for Atp2a2 and one shRNA for PPAR α caused a significant reduction of the target proteins. In Fig S6c, only one PPAR α overexpression clone exhibited a significant increase of PPAR α expression. The authors need to repeat the functional assays with an independent siRNA/shRNA or an independent PPAR α overexpression clone.

Response: Thanks so much for this insightful and excellent comment and suggestion. We do agree with the reviewer's point that repeating the functional assays with another independent siRNA/shRNA is really necessary. Thus, in addition to the siRNA/shRNA used in our original manuscript, we have generated another independent siRNA for Atp2a2 and another independent shRNA for PPAR α to suppress the expression of the target proteins. We have confirmed the knockdown efficacy of these newly constructed siRNA and shRNA, and have also repeated the relevant functional assays. These new data and the corresponding description have been added into the revised manuscript (see Figure 4, Figure 9, Figure S7, Figure S8, Figure S18, Results L233, Supplemental Table 4). Moreover, it might be impossible to construct another independent PPAR α overexpression clone, because cDNA encoding the full-length PPAR α gene was amplified to construct the lentiviral vector for PPAR α overexpression. In our original manuscript, the overexpression lentivirus was constructed by three different companies using different promoters, while we selected the PPAR α overexpression clone that exhibited the most remarkable increase of the PPAR α expression to perform the functional assays. However, we do realize that this western blotting image might provide potentially confusing and misleading information for the readers. Thus, we have performed western blotting assays again only using this valid PPAR α overexpression vector, and the corresponding western blotting results have been replaced in the revised manuscript (see Figure S7).

5. The FSS/loading-induced calcium oscillation in osteocytes is maintained by both SERCA2 and IP₃R as shown by the authors. Unfortunately, the potential involvement of IP₃R in T2D-induced impairment of calcium oscillation in osteocytes and deterioration of bone mechanosensitivity has not been properly addressed. Please clarify whether the P₂R-PLC-IP₃-IP₃R pathway is altered in osteocytes in diabetes or not.

Response: Many thanks for this professional and excellent suggestion and comment. Firstly, as suggested, we have supplemented extra experiments to determine Ca²⁺ oscillatory dynamics in osteocytes in response to fluid shear stress (FSS) stimulation following the inhibition of IP₃R. Our new results showed that treatment with the IP₃R antagonist significantly inhibited FSS-induced Ca²⁺ oscillations in normal MLO-Y4 cells and totally abolished the FSS-induced Ca²⁺ oscillations in HGHF-treated MLO-Y4 cells, which was similar to the findings of treatment with the P₂R or PLC antagonist as shown in Figure S5 in our original manuscript. These new data associated with the IP₃R antagonist and the corresponding description have been added into the revised manuscript (see Figure S5, Results L218-L219, and Supplemental Information L80). Secondly, we have also supplemented extra western blotting experiments to assess whether the purinergic P₂Y receptor-mediated PLC-IP₃-IP₃R pathway is altered in osteocytes in diabetes according to the reviewer's suggestion. These new western blotting results showed no significant difference was observed in the protein expression of P₂Y₂, P₂Y₄, and P₂Y₁₂ (three major functional isoforms expressed in osteocytes), PLCβ1 (P₂Y-downstream main effector isoform), or IP₃R between normal and HGHF-treated MLO-Y4 cells. These new data and the corresponding description have been added into the revised manuscript (see Figure S6, Results L229-L232, Methods L776-L778, and Supplemental Information L90-L93). Our findings reveal that both P₂Y-PLC-IP₃-IP₃R and SERCA are important for the maintenance of Ca²⁺ dynamics in T2D (P₂Y-PLC-IP₃-IP₃R is responsible for Ca²⁺ release, and SERCA is responsible for Ca²⁺ reuptake), whereas T2D-induced decrease in the expression of SERCA2 rather than P₂Y-PLC-IP₃-IP₃R mediates the suppression of osteocytic Ca²⁺ oscillatory dynamics and bone mechano-responsiveness in T2D.

Major points:

1. The western blots should be quantified across the MS.

Response: Thanks so much for this insightful comment. Actually, all the images of western blots in the main figures have their corresponding quantification and statistical analysis results either in the main figures or in the supplemental figures (western blots in Figure 4d have the corresponding statistical data in this figure, the corresponding statistical data of Figure 5f is in Figure S11c, the statistical data of Figure 5g is in Figure S11e, the statistical data of Figure 7i is in Figure S15b, the statistical data of Figure 7m is in Figure S15d, western blots in Figure 9d have the corresponding statistical data in this figure, the statistical data of Figure 9f is in Figure S18a, the statistical data of Figure 9g is in Figure S18b, the statistical data of Figure 9m is in Figure S18e, and the statistical data of Figure 9o is in Figure S18h).

2. The experiments were only performed on male mice. It would be nice to show whether this mechanism is also critical in female mice.

Response: Many thanks for this insightful and valuable suggestion and comment. Most of the current animal studies for T2D and the relevant complications used male rodents, owing to the protective role of estrogens against the development of the metabolic dysfunctions [1-3]. However, we do agree with the reviewer's point that the data in female mice are also interesting and important. Considering that female pancreatic islets are much less sensitive to STZ-induced cytotoxicity than male mice [4-5], we supplemented extra experiments on genetically spontaneous female KK-Ay mice. Our findings showed that the osteocytic Ca^{2+} oscillatory response to mechanical loading was also observed to be reduced in female KK-Ay mice. A significant decrease in the SERCA2 expression in osteocytes was also found in female KK-Ay mice compared to controls. Moreover, increasing SERCA2 expression by the treatment with ISTA was observed to enhance Ca^{2+} oscillatory responses to mechanical loading in osteocytes in female KK-Ay mice. The relevant female data and the corresponding description have been added into the revised manuscript (see Figure S10, Results L260-L263, Methods L549, and Supplemental Information L127-L137).

References

- [1] Paik SG, Michelis MA, Kim YT, Shin S. Induction of insulin-dependent diabetes by streptozotocin. Inhibition by estrogens and potentiation by androgens. *Diabetes*. 1982 Aug;31(8 Pt 1):724-9.
- [2] Shin YK, Hsieh YS, Han AY, Kwon S, Kang P, Seol GH. Sex-specific susceptibility to

type 2 diabetes mellitus and preventive effect of linalyl acetate. Life Sci. 2020 Nov 1;260:118432.

[3] Kim B., Kim YY, Nguyen PTT, Nam H, Suh JG. Sex differences in glucose metabolism of streptozotocin-induced diabetes inbred mice (C57BL/6J). *Appl Biol Chem.* 2020 Sept 22; 63:59.

[4] Kolb H. Mouse models of insulin dependent diabetes: low-dose streptozocin-induced diabetes and nonobese diabetic (NOD) mice. *Diabetes Metab Rev.* 1987 Jul;3(3):751-78.

[5] Furman BL. Streptozotocin-Induced Diabetic Models in Mice and Rats. *Curr Protoc.* 2021 Apr;1(4):e78.

3. The putative PPAR α binding sites were found in *Atp2a2* promoter. The authors need to demonstrate whether PPAR α indeed binds to these sites via a gel shift experiment.

Response: Thanks so much for this insightful and professional suggestion. We do agree with the reviewer's point that the electrophoretic mobility shift assays (EMSA) are necessary for our current study. Thus, as suggested, we have supplemented the EMSA experiment during the manuscript revision period. Our new EMSA findings provide further evidence to confirm that PPAR α indeed binds to the *Atp2a2* promoter region (-620 to -608 bp) instead of the promoter region (-1283 to -1277 bp), and the binding activity was significantly decreased under the HGHF condition. The relevant data and the corresponding description have been added into the revised manuscript (see Figure 9I, Results L418-L422, Methods L817-L825, Figure Legends L1192-L1197).

4. In Fig 8g-i, the authors indicate $n=6$. Please clarify whether this means images from 6 mice or 6 images from one mouse. They need to clarify such information in many other figures throughout the MS.

Response: The statement $n=6$ in Figure 8g-k indicates images collected from six mice. We have clarified all such information throughout the Figure Legends section of our manuscript. It is really a professional and insightful comment and suggestion, which is helpful for making the statement of the manuscript much clearer.

5. On Page 19, the author wrote that 'we found that the T2D-induced reduction of bone mechanosensitivity was attributed to the increased phosphorylation of PPAR α -mediated specific reduction in the expression of osteocytic SERCA2'. They need to present the data on phosphorylation of PPAR α .

Response: Many thanks for this professional and excellent suggestion and comment. As suggested, we have performed extra western blotting experiments to assess the phosphorylation of PPAR α in T2D osteocytes. These new western blotting results and the corresponding descriptions have been added into the revised manuscript (see Figure 9d, Results L394-396, and Figure Legends L1178-L1179).

Reviewer #2

Shao et al., describe the role of SERCA2 in the lack of anabolic response of osteocytes to error loading. It is an elegant study, combining validated in vivo models with well-established cell culture experiments in order to distinguish between systemic effects of T2D on bone vs. direct effects on osteocytes. En passant the study clearly demonstrates the importance of including mechanical stimuli in bone research (e.g. fig 6 ISTA does not affect bone parameters in the absence of mechanical stimulation). Overall a substantial amount of work has been performed, leading to an accumulation of convincing results. The results are interesting for a large audience, and are likely to lead to societal impact.

I did not find a specifically formulated goal or hypothesis. This is unfortunate, because I wonder whether the proper loading regime was applied.

The tibia loading model creates a positive error load in one tibia, that exceeds the daily mechanical loads generated by the physical activity of, in this case, mice going about their business. The control tibia is thus not “unloaded”, but rather “normally loaded”. An error load creates a gain in bone mass in the loaded bone, as nicely shown in several figures in the current manuscript. The error load can be compared to a human whom will take up jogging (at an easy pace), but may not represent normal activity. Thus, the lack of anabolic response in loaded tibia in T2D mouse models indicate that exercise will not benefit bone properties in people with T2D, but it does not necessarily explain the occurrence of the bone phenotype in T2D.

Istaroxime was effective in loaded bones (people who take up jogging), but did not affect the control bones (people who go about their normal daily activities). Thus, following the paper, potential therapeutic effects should be based on a combination of an exercise regime and taking SERCA2 agonists.

Experiments in which hindlimbs of T2D mice are unloaded and bone parameters compared to those of normal ambulatory mice would generate significant additional insights. Without such experiments the introduction needs some rewriting to avoid the suggestion that the etiology of bone fragility in normal ambulatory T2D patients is investigated.

Response: Many thanks for these professional and insightful comments and suggestions. Our response for these concerns is as follows.

Firstly, we do agree with the reviewer’s point that a specifically formulated goal or hypothesis is indeed necessary in the last paragraph of the Introduction section, although the relevant statement was missing in our original manuscript. Thus, a specific scientific hypothesis has been added into the revised manuscript according to the suggestion (see Introduction L97-L98).

Secondly, our group has carefully discussed about the reasonability for naming the right tibia “nonloaded”, and we do realize that this statement is really not appropriate enough, since the right hindlimb can also bear daily normal ambulatory loading. Thus, the statement “nonloaded” has been replaced by “control” in the revised manuscript (see Figure 1, Figure 2, Figure 6, Figure 7, Figure 8, Figure S1, Figure S13, Figure S14, and Figure S17).

Thirdly, we do agree with the reviewer’s point that potential therapeutic strategy should be based on a combination of an exercise regime (rather than normal daily activities) and taking SERCA2 agonists according to our findings. Thus, the relevant sentences have been modified to provide a more precise and appropriate statement in the revised manuscript (see Discussion L535).

Finally, it is indeed true just as mentioned by the reviewer that our findings indicate that exercise will not benefit bone properties in T2D, whereas it does not necessarily explain the occurrence of the bone phenotype in T2D. Thus, we have made revisions throughout the Abstract, Introduction and Discussion sections of our manuscript to avoid the statement that the etiology of bone fragility in normal ambulatory T2D patients is investigated. It is really a professional and insightful suggestion, which is helpful for making the interpretation of our results more precise and convincing.

The first paragraph of the introduction is suggestive that people with T2D have less bone, similar to people who experience unloading for prolonged periods of time. However, most studies do not find a reduction in BMD in T2D in humans. On the contrary, increases in BMD in T2D have been regularly reported, although cortical porosity also clearly increases (as describes in ref 5 of the manuscript). A recent meta analysis (Walle et al., current osteoporosis rep 2022) even goes so far as to claim that T2DM effects were more severe at the radius than the load-bearing tibia, supporting mechanoregulation of bone fragility in diabetes mellitus.

Response: Thanks so much for this excellent and valuable comment. Firstly, our original manuscript stated that “In contrast to non-diabetics, significant declines in cortical thickness and trabecular and cortical material properties, and increases in cortical porosity have been observed in T2D patients”. However, we have carefully gone through the literatures and found that the cortical thickness data in T2D patients in contrast to non-diabetes are inconsistent and controversial. Thus, the statement “significant declines in cortical thickness” in T2D has been

removed in the revised manuscript (see Introduction L47-L48).

Secondly, the well-designed meta-analysis paper reported by Walle et al. (Current Osteoporosis Rep, 2022) provides a systemic summarization for T2D-associated differences in bone structure. According to the analysis of this paper, cortical porosity was more severe at the radius than the tibia in T2D patients, although many other parameters (including Tt.Ar, Tb.BMD, Ct.BMD, Tb.N, Tb.Sp, Tb.Th, Tb.1/N.SD, Ct.Th, and FL) exhibited no significant difference between the radius and tibia. This difference in cortical architecture between the radius and tibia may be associated with many various factors in addition to mechanical loading, such as hormones, life styles, and age [1-4]. Moreover, existing data regarding the difference of bone architecture between the radius and tibia in T2D remain diverse and inconsistent [5-10], and further extensive clinical trials are still needed to verify the summarization and prediction results of this elaborate meta-analysis paper. Furthermore, no direct evidence can be provided for the comparison of bone mechano-responsiveness at a specific identical site between normal and T2D to date either by this meta-analysis paper or any other studies. Our current study represents the first effort to systematically compare the mechano-responsiveness between normal and T2D skeletons by using two typical T2D animal models (i.e., genetically spontaneous KK-Ay mice and experimentally-induced HFD+STZ mice), and provides direct evidence to reveal that tibial mechano-responsiveness in T2D animals is significantly lower than that in non-diabetic controls. We do realize that the paper by Walle et al. is really important for our paper, and thus add it into the reference list of the revised manuscript (see Reference #6).

References

- [1] Walsh JS, Paggiosi MA, Eastell R. Cortical consolidation of the radius and tibia in young men and women. *J Clin Endocrinol Metab.* 2012 Sep;97(9):3342-8.
- [2] Paccou J, Edwards MH, Ward K, Jameson K, Moon R, Dennison E, Cooper C. Relationships between bone geometry, volumetric bone mineral density and bone microarchitecture of the distal radius and tibia with alcohol consumption. *Bone.* 2015 Sep;78:122-9.
- [3] Burghardt AJ, Kazakia GJ, Ramachandran S, Link TM, Majumdar S. Age- and gender-related differences in the geometric properties and biomechanical significance of intracortical porosity in the distal radius and tibia. *J Bone Miner Res.* 2010 May;25(5):983-93.

- [4] Sode M, Burghardt AJ, Kazakia GJ, Link TM, Majumdar S. Regional variations of gender-specific and age-related differences in trabecular bone structure of the distal radius and tibia. *Bone*. 2010 Jun;46(6):1652-60.
- [5] Shanbhogue VV, Hansen S, Frost M, Jørgensen NR, Hermann AP, Henriksen JE, Brixen K. Compromised cortical bone compartment in type 2 diabetes mellitus patients with microvascular disease. *Eur J Endocrinol*. 2016 Feb;174(2):115-24.
- [6] Burghardt AJ, Issever AS, Schwartz AV, Davis KA, Masharani U, Majumdar S, Link TM. High-resolution peripheral quantitative computed tomographic imaging of cortical and trabecular bone microarchitecture in patients with type 2 diabetes mellitus. *J Clin Endocrinol Metab*. 2010 Nov;95(11):5045-55.
- [7] Paccou J, Ward KA, Jameson KA, Dennison EM, Cooper C, Edwards MH. Bone Microarchitecture in Men and Women with Diabetes: The Importance of Cortical Porosity. *Calcif Tissue Int*. 2016 May;98(5):465-73.
- [8] Nilsson AG, Sundh D, Johansson L, Nilsson M, Mellström D, Rudäng R, Zoulakis M, Wallander M, Darelid A, Lorentzon M. Type 2 Diabetes Mellitus Is Associated With Better Bone Microarchitecture But Lower Bone Material Strength and Poorer Physical Function in Elderly Women: A Population-Based Study. *J Bone Miner Res*. 2017 May;32(5):1062-1071.
- [9] Samelson EJ, Demissie S, Cupples LA, Zhang X, Xu H, Liu CT, Boyd SK, McLean RR, Broe KE, Kiel DP, Bouxsein ML. Diabetes and Deficits in Cortical Bone Density, Microarchitecture, and Bone Size: Framingham HR-pQCT Study. *J Bone Miner Res*. 2018 Jan;33(1):54-62.
- [10] Farr JN, Drake MT, Amin S, Melton LJ 3rd, McCready LK, Khosla S. In vivo assessment of bone quality in postmenopausal women with type 2 diabetes. *J Bone Miner Res*. 2014 Apr;29(4):787-95.

Also, unloading is associated with an increased turnover with a negative balance resulting on bone loss (as observed in the current study in fig 2 for the T2D models) while T2D in humans is generally associated with decreased turnover, resulting a phenotype mimicking advanced skeletal ageing. I missed the link with the ageing phenotype and increased senescence in the current paper.

Response: It is really a professional and insightful comment. It is well established that both T2D humans and animals are associated with the suppressed bone formation, as characterized by significant decreases in serum osteocalcin, P1NP, and histological numbers of osteoblasts [1-2]. However, the effects of T2D on bone resorption are less consistently described, and the relevant literatures remain controversial. In humans, most of bone resorption markers (e.g., CTX-1 and TRAcP5b) are shown by most researchers to be reduced in T2DM individuals, while others

reveal a significant increase (e.g., CTX-1 and NTX) [3-4]. In animals, however, bone resorption parameters are mostly increased (serum CTX-1, serum TRAcP5b, or histological numbers of osteoclasts) in most T2D rodent models (e.g., MKR mice, db/db mice and ZDF rats) [5-9]. Previous studies have also shown a significant increase in bone resorption in both KK-Ay and HFD+STZ mice [10-15], which were consistent with our current findings. The lack of an ideal animal model that can precisely mimic the skeletal phenotype in T2D patients is a major and unavoidable issue encountered by researchers for the study of the T2D effects on bone physiology [16-17]. Thanks so much for this valuable comment raised by the Reviewer, and we have added the relevant discussion for this concern into the Discussion section of the revised manuscript (see Discussion L454-L458).

References

- [1] Compston J. Type 2 diabetes mellitus and bone. *J Intern Med.* 2018 Feb;283(2):140-153. doi: 10.1111/joim.12725. Epub 2018 Jan 8.
- [2] Song F, Lee WD, Marmo T, Ji X, Song C, Liao X, Seeley R, Yao L, Liu H, Long F. Osteoblast-intrinsic defect in glucose metabolism impairs bone formation in type II diabetic male mice. *Elife.* 2023 May 5;12:e85714.
- [3] Faienza MF, Pontrelli P, Brunetti G. Type 2 diabetes and bone fragility in children and adults. *World J Diabetes.* 2022 Nov 15;13(11):900-911.
- [4] Takizawa M, Suzuki K, Matsubayashi T, Kikuyama M, Suzuki H, Takahashi K, Katsuta H, Mitsuhashi J, Nishida S, Yamaguchi S, Yoshimoto K, Itagaki E, Ishida H. Increased bone resorption may play a crucial role in the occurrence of osteopenia in patients with type 2 diabetes: Possible involvement of accelerated polyol pathway in its pathogenesis. *Diabetes Res Clin Pract.* 2008 Oct;82(1):119-26.
- [5] Kawashima Y, Fritton JC, Yakar S, Epstein S, Schaffler MB, Jepsen KJ, LeRoith D. Type 2 diabetic mice demonstrate slender long bones with increased fragility secondary to increased osteoclastogenesis. *Bone.* 2009 Apr;44(4):648-55.
- [6] Catalfamo DL, Britten TM, Storch DL, Calderon NL, Sorenson HL, Wallet SM. Hyperglycemia induced and intrinsic alterations in type 2 diabetes-derived osteoclast function. *Oral Dis.* 2013 Apr;19(3):303-12.
- [7] Jing D, Luo E, Cai J, Tong S, Zhai M, Shen G, Wang X, Luo Z. Mechanical Vibration Mitigates the Decrease of Bone Quantity and Bone Quality of Leptin Receptor-Deficient Db/Db Mice by Promoting Bone Formation and Inhibiting Bone Resorption. *J Bone Miner Res.* 2016 Sep;31(9):1713-24.
- [8] Liu R, Bal HS, Desta T, Krothapalli N, Alyassi M, Luan Q, Graves DT. Diabetes enhances periodontal bone loss through enhanced resorption and diminished bone

formation. *J Dent Res.* 2006 Jun;85(6):510-4.

[9] Picke AK, Gordaliza Alaguero I, Campbell GM, Glüer CC, Salbach-Hirsch J, Rauner M, Hofbauer LC, Hofbauer C. Bone defect regeneration and cortical bone parameters of type 2 diabetic rats are improved by insulin therapy. *Bone.* 2016 Jan;82:108-15.

[10] Fu C, Zhang X, Ye F, Yang J. High insulin levels in KK-Ay diabetic mice cause increased cortical bone mass and impaired trabecular micro-structure. *Int J Mol Sci.* 2015 Apr 13;16(4):8213-26.

[11] Shao X, Yang Y, Tan Z, Ding Y, Luo E, Jing D, Cai J. Amelioration of bone fragility by pulsed electromagnetic fields in type 2 diabetic KK-Ay mice involving Wnt/ β -catenin signaling. *Am J Physiol Endocrinol Metab.* 2021 May 1;320(5):E951-E966.

[12] Guo Y, Wang L, Ma R, Mu Q, Yu N, Zhang Y, Tang Y, Li Y, Jiang G, Zhao D, Mo F, Gao S, Yang M, Kan F, Ma Q, Fu M, Zhang D. JiangTang XiaoKe granule attenuates cathepsin K expression and improves IGF-1 expression in the bone of high fat diet induced KK-Ay diabetic mice. *Life Sci.* 2016 Mar 1;148:24-30.

[13] Eckhardt BA, Rowsey JL, Thicke BS, Fraser DG, O'Grady KL, Bondar OP, Hines JM, Singh RJ, Thoreson AR, Rakshit K, Lagnado AB, Passos JF, Vella A, Matveyenko AV, Khosla S, Monroe DG, Farr JN. Accelerated osteocyte senescence and skeletal fragility in mice with type 2 diabetes. *JCI Insight.* 2020 May 7;5(9):e135236.

[14] Behera J, Ison J, Voor MJ, Tyagi SC, Tyagi N. Diabetic Covid-19 severity: Impaired glucose tolerance and pathologic bone loss. *Biochem Biophys Res Commun.* 2022 Sep 10;620:180-187.

[15] Ham JR, Choi RY, Lee HI, Lee MK. Methoxsalen and Bergapten Prevent Diabetes-Induced Osteoporosis by the Suppression of Osteoclastogenic Gene Expression in Mice. *Int J Mol Sci.* 2019 Mar 14;20(6):1298.

[16] Fajardo RJ, Karim L, Calley VI, Bouxsein ML. A review of rodent models of type 2 diabetic skeletal fragility. *J Bone Miner Res.* 2014;29(5):1025-40.

[17] Hamann C, Goettsch C, Mettelsiefen J, Henkenjohann V, Rauner M, Hempel U, Bernhardt R, Fratzl-Zelman N, Roschger P, Rammelt S, Günther KP, Hofbauer LC. Delayed bone regeneration and low bone mass in a rat model of insulin-resistant type 2 diabetes mellitus is due to impaired osteoblast function. *Am J Physiol Endocrinol Metab.* 2011 Dec;301(6):E1220-8.

The authors regularly mention “mechanosensitivity”, suggesting that the setpoint of the mechanostat in osteocytes is changed with T2D, whereby osteocytes need a different intensity stimulus to respond. This is not what has been investigated. The magnitude of the response to 1 type of load has been investigated, which corresponds to a change in “mechanoresponsiveness”. In the abstract both words are used.

Response: Thanks so much for this insightful and valuable comment. We do agree with the reviewer’s point that the words “mechanosensitivity”

and “mechano-responsiveness” may not confer exactly the same meaning, and the word “mechano-responsiveness” may be a more appropriate expression used for our manuscript. Thus, the two words have been unified into the word “mechano-responsiveness” throughout our revised manuscript.

The materials and methods seem sound. The number of independent experiments should be mentioned. Why did the authors used a diamond saw to make sections rather than a microtome?

Response: Many thanks for this excellent and professional suggestion and comment. Firstly, as suggested, we have specified the sample number or the number of biologically independent replicates for each assay in the Figure Legends section throughout the revised manuscript. Secondly, all the undecalcified bone specimens in the current study were sectioned using a Leica SpA diamond saw microtome (i.e., a commercial microtome equipped with diamond-coated saw blades). We have replaced the statement “diamond saw microtome” with “a diamond saw in a microtome” in the revised manuscript to make the statement much clearer (see Methods L624).

Sentence 70 is difficult to follow.

Response: Thanks a lot for this professional and valuable comment. We have rewritten the whole sentence to avoid providing any confusing and imprecise information (see Introduction L65-L67).

Sentence 55: Vertebra are not part of the lower extremities.

Response: The word “vertebra” has been removed in the revised manuscript (see Introduction L51). Many thanks for this professional comment.

Reviewer #3

The current manuscript submitted by Shao and colleagues, show here that Calcium dynamics is important for the mechanosensitive function of osteocytes during mechanical loading. Mechanical loading failed to improve osteocyte function or bone parameters in models of Type2 diabetes due to lack of response to calcium signaling as a result of downregulation of SERCA2. Serca2 antagonists suppressed Calcium signaling in osteocytes, while Serca2 agonists, decrease cortical porosity, improved strength, and improved net bone formation. The study is well designed, and data mostly supports the hypothesis that SERCA2 is crucial for osteocyte mechanosensitivity to loading. However, there are certain deficiencies that are pointed out here.

1. Please quantify adipocyte numbers and volume per bone volume since Figure 1e suggest there are changes in adipocytes.

Response: Many thanks for this professional and insightful suggestion and comment. We do agree with the reviewer's point that the data of changes in adipocytes are also necessary for our current study. Thus, the data of the adipocyte numbers and volume per bone volume and the corresponding description have been added into the revised manuscript (see Figure S1, Results L135-L137, Methods L641-L642, and Supplemental Information L32-L33).

2. The data in Figure 2a, showing empty lacuna in KK-Ay and HFD-STZ mice does not match with staining patterns in 2b, 2c, 2d and 2e where the empty lacuna is not seen. How is it possible that one sample from an animal has empty lacuna, while others do not. Furthermore, these empty lacunae are not seen in Figures 4f, 6f, 6g, 6h and 6i. This is not consistent with what the authors have shown in 2a. Please explain these discrepancies.

Response: Thanks so much for this professional and insightful comment. According to the statistical data of empty lacuna based on H&E staining as shown in Figure 2a in the original manuscript (Figure S1g in the revised manuscript), the percentage of empty lacuna was approximately 30% in both KK-Ay and HFD+STZ mice. That is, most of osteocytes (60%~70%) remain alive in these two T2D animal models. In our original manuscript, in order to better show the visual difference in the number of positive cells between normal and T2D animals based on immunohistochemical staining, we specially selected the field of view with less empty lacunae (i.e., more healthy osteocytes) as the representative immunohistochemical images (but a few empty lacunae can still be seen in these immunohistochemical images). However, we do

realize that the inconsistency in the percentage of empty lacuna between our representative immunohistochemical images and the H&E staining data in our original manuscript really provides quite confusing information for the readers. Thus, almost all of the representative immunohistochemical images have been replaced by new images in the revised manuscript to make sure that the percentage of empty lacunae shown in these representative immunohistochemical images kept consistent with the H&E staining results (see Figure 2, Figure 4, Figure 6, Figure 7, Figure 8, Figure S13, Figure S14, Figure S16, and Figure S17).

3. What about osteocyte empty lacunae data following ISTA treatment in KK-Ay mice? That should be presented.

Response: Many thanks for this professional and valuable suggestion. We do agree with the reviewer's point that the empty lacunae data following ISTA treatment in HFD+STZ mice are really necessary for our study. Thus, the relevant data and the corresponding description have been added into the revised manuscript (see Figure S14, Results L326-L328, and Supplemental Information L177-L178).

4. Osteocyte canalicular network is a key measure of osteocyte health and has direct indication for mechanosensation. Hence this makes canalicular networks as a key piece of data which should be shown in both models of Typ2 diabetes and following SERCA2 agonist treatment.

Response: We do agree with the reviewer's point that osteocyte canalicular network is important for reflecting osteocyte health and also has direct indication for bone mechano-responsiveness. Thus, we have supplemented the Ploton silver staining to visualize the morphology of osteocyte canalicular network based on the decalcified bone sections. These new osteocyte canalicular network data have been added into the revised manuscript (see Figure 2, Figure S1, Figure S13, Figure S14, and Figure S17). The corresponding description for these new data has also been added (see Results L151-L153, L157, L306-L307, L326-L328, L370, Methods L643-L645, Figure Legends L1036-L1038, and Supplemental Information L36-L37, L165-L166, L179-L180, L213-L214).

5. Dynamic histomorphometry was shown in the paper, however static histomorphometry of osteoblast numbers was missing. That should be shown as well. This will support the data from dynamic histomorphometry.

Response: Many thanks for this insightful and valuable suggestion and

comment. Actually, static bone histomorphometric analysis of osteoblast numbers had been performed based on toluidine blue staining, but most of the corresponding toluidine blue staining data were present in the supplemental figures in our original manuscript. However, we do agree with the reviewer's point that data of static histomorphometry of osteoblast numbers are quite important. Thus, these data have been moved to the main figures in the revised manuscript (see Figure 6j, Figure 7g, and Figure 8j). The corresponding description has also been revised in the Figure Legends section (see Figure Legends L1121-L1122, L1136-L1137, L1165-L1167).

6. One key data missing in all experimental design was markers for bone formation (PINP), resorption (CTX-I) and sclerostin in the serum. This should be presented.

Response: We do agree with the reviewer's point that serum P1NP, CTX-1 and sclerostin are important markers those can provide additionally important information about bone turnover and bone cell functions. Thus, we have supplemented an extra determination for the levels of these markers in the serum samples that have been stored at -80°C using the commercial ELISA kits. The corresponding results have been added into the revised manuscript (see Figure S1, Figure S13, Figure S14, and Figure S17). The relevant descriptions for these serum results and the corresponding methodology have been also added into the revised manuscript (see Results L168-L171, L308-L310, L333-L335, L375-L377, and Methods L659-L663, Supplemental Information L37-L38, L166-L167, L180-L181, L214-L215).

REVIEWER COMMENTS

Reviewer #1 (Remarks to the Author):

The authors have rectified some errors in the original manuscript, and also adequately addressed my concerns. No further comments.

Reviewer #2 (Remarks to the Author):

The revisions have significantly improved the manuscript and I am happy with the responses given.

The addition of visualization of the canalicular network in figures 2, S1, S13, S14 and S17 particularly strengthen the manuscript, since a difference in response to mechanical loading of bone in vivo might indeed originate from changes in osteocyte shape, size, or alterations in the canalicular network (e.g. density) at baseline (before start of the loading regime).

Interestingly, none of the baseline values in canalicular length or density differs between the groups, suggesting that alterations in lacuno-canalicular architecture can not explain the different response to loading in the mice with T2D. It is recommended to explicitly mention this in the discussion.

It would be good to double check the statistical results first though, since it is striking that there is a significant effect of loading on canalicular density in non-diabetes mice, but no difference between any of the controls (fig S1h). Is it possible that certain significant results (at baseline) have not been shown for any reason?

Also it is noteworthy that canalicular length and density are altered within a 2 week timeframe (duration of the loading regime) in all figures. It means that either all the bone has been replaced by new bone containing new osteocytes with more extensions, or new canaliculi have been created by the existing osteocytes in the existing calcified matrix in this two weeks timeframe. Theoretically this is not impossible, but it is noteworthy. This response to loading is unknown to me and I was not able to find an other publication describing this observation during a quick search. If the authors can provide 1 or two references describing this response of the canalicular network to loading it would significantly strengthen this observation as outcome measure.

Reviewer #3 (Remarks to the Author):

The authors have done a decent job in addressing the concerns I raised. I have no further questions.

Response to Reviewers

Manuscript No. NCOMMS-22-46839A

Below are our responses (in BOLD type) to the Referee's comments on a point-by-point basis.

Reviewer #2 (Remarks to the Author):

The revisions have significantly improved the manuscript and I am happy with the responses given.

The addition of visualization of the canalicular network in figures 2, S1, S13, S14 and S17 particularly strengthen the manuscript, since a difference in response to mechanical loading of bone in vivo might indeed originate from changes in osteocyte shape, size, or alterations in the canalicular network (e.g. density) at baseline (before start of the loading regime).

Interestingly, none of the baseline values in canalicular length or density differs between the groups, suggesting that alterations in lacuno-canalicular architecture can not explain the different response to loading in the mice with T2D. It is recommended to explicitly mention this in the discussion. It would be good to double check the statistical results first though, since it is striking that there is a significant effect of loading on canalicular density in non-diabetes mice, but no difference between any of the controls (fig S1h). Is it possible that certain significant results (at baseline) have not been shown for any reason?

Response: Thanks so much for this insightful and excellent comment and suggestion. In our experiment, the left tibiae of mice were subjected to cyclic compressive loading for 2 weeks, and the contralateral right tibiae were not loaded and served as controls. In the figures of our original manuscript, we did not mark statistical differences (with asterisks) in the non-load control side between T2D and non-diabetic mice, because the main focus was originally concentrated on the comparisons of the loading effect between T2D and non-diabetic mice. However, there are actually significant differences in the canalicular length and density in the non-load control side between T2D and non-diabetic mice ($P<0.001$). We do agree with the reviewer's point that comparisons in the non-load side between T2D and non-diabetic mice are also important. Thus, we have provided marks with asterisks for the comparisons in the non-load control sides those have statistical difference in the figures throughout the manuscript. The corresponding description for this important concern has been added into the Results

section of the revised manuscript (see Results L151-L155). It is really a professional and valuable comment, which is helpful for improving the quality of our manuscript.

Also it is noteworthy that canalicular length and density are altered within a 2 week timeframe (duration of the loading regime) in all figures. It means that either all the bone has been replaced by new bone containing new osteocytes with more extensions, or new canaliculi have been created by the existing osteocytes in the existing calcified matrix in this two weeks timeframe. Theoretically this is not impossible, but it is noteworthy. This response to loading is unknown to me and I was not able to find an other publication describing this observation during a quick search. If the authors can provide 1 or two references describing this response of the canalicular network to loading it would significantly strengthen this observation as outcome measure.

Response: Many thanks for this insightful and professional comment and suggestion. Actually, the positive effects of mechanical loading on the canalicular network in long bone have been documented by previous animal studies [1-3]. Similarly, some *in vitro* studies have also reported that fluid shear stress induced a significant increase in osteocyte dendrites [4-5].

In Reference #1, Aido MIFd found that uniaxial cyclic compressive loading for 2 weeks induced a significant increase in canalicular density when compared with the contralateral control tibiae (Table 6 in page 82 in the thesis). Similarly, in Reference #2, Schemenz V reported that, as compared with the control tibiae, mechanically-loaded tibiae (uniaxial cyclic compressive loading for 2 weeks) exhibited a significant increase in canalicular density (Table 5.2 in page 52 and Figure 5.9 in page 55 in the thesis). In Reference #3, Li et al. found that mice subjected to bone-loading exercise via treadmill training for 6 weeks showed a significant increase in osteocyte dendrite (canalicular) length and density compared with the control mice those were not subjected to treadmill exercise (Figure 4E&F in the paper).

However, we do realize that Reference #1 and #2 are PhD theses rather than peer-reviewed published articles, and the timeframe of mechanical loading in Reference #3 is 6 weeks rather than 2 weeks. Thus, we have collaborated with another lab group (Professor Bin Wang's lab from Chongqing Medical University in China) to replicate the same experiment independently according to the suggestion from the editor and reviewer. Professor Bin Wang has long focused on the remodeling of lacunar-canalicular system under mechanical loading. Using the same

experimental protocol, their results show that uniaxial cyclic compressive loading on the tibia for 2 weeks with the peak strain of 1455 $\mu\epsilon$ resulted in a significant increase in canalicular length and density in 3-month-old male mice as compared with the contralateral control tibia, which are consistent with our current findings. The detailed description for the experimental methods and results performed by Professor Bin Wang's lab can be seen in the appendix in the end of this letter. We have also acknowledged their contribution and kind help in the Acknowledgement section of the revised manuscript (see Acknowledgements L1004-L1007).

References

- [1] Aido MIFd. *The influence of age and mechanical loading on bone structure and material properties*. Technische Universität, Berlin, 2015.
- [2] Schemenz V. *Correlations between osteocyte lacuno-canalicular network and material characteristics in bone adaptation and regeneration*. Universität Potsdam, 2022.
- [3] Li Q, Wang R, Zhang Z, et al. *Sirt3 mediates the benefits of exercise on bone in aged mice*. *Cell Death Differ*. 2023;30(1):152-167.
- [4] Wang W, Sarazin BA, Kornilowicz G, et al. *Mechanically-Loaded Breast Cancer Cells Modify Osteocyte Mechanosensitivity by Secreting Factors That Increase Osteocyte Dendrite Formation and Downstream Resorption*. *Front Endocrinol (Lausanne)*. 2018;9:352.
- [5] Zhang K, Barragan-Adjemian C, Ye L, et al. *E11/gp38 selective expression in osteocytes: regulation by mechanical strain and role in dendrite elongation*. *Mol Cell Biol*. 2006;26(12):4539-52.

APPENDIX

The experiment was performed by Xingfan Wu and Bin Wang, from the Institute of Life Sciences, College of Basic Medicine, Chongqing Medical University, Chongqing 400016, China.

Correspondence email: nc.ude.umqc@gnawb (Bin Wang)

Materials and Methods

Animals

Male C57BL/6 mice (12 weeks old) were purchased from Chongqing Medical University's Animal Center. Mice were bred under a 12-h light-dark cycle with ambient temperature of 22~24°C and relative humidity of 55 ± 5%. All animal experiments were approved by the Committees of Animal Ethics and Experimental Safety of Chongqing Medical University.

In vivo mechanical loading

The relationship of force and strain was measured on the tibiae of 12-week-old male mice. The strain gauge (EA-06-015DJ-120, Measurements Group Inc., NC, USA) was bonded to the anteromedial surface of the tibia. The tibia was subjected to axial compressive loading ranging from 0 to 12 N. The peak load of 8.85 N corresponding to 1455 µε strain on tibial surfaces were obtained in the C57BL/6 mice. After anesthetization by isoflurane, the left tibiae of mice were subjected to cyclic compressive loading in a ramp loading waveform at 4 Hz with magnitude of 8.85 N. The loading was performed at 1200 cycles/day for 5 consecutive days/week for 2 weeks. The contralateral non-loaded limbs (right tibiae) were served as internal controls. Mice were euthanized and bilateral tibiae were dissected and harvested after final loading episode.

Ploton silver staining

Tibial specimens were fixed in 4% formaldehyde and immersed in the decalcifying solution. Samples were dehydrated through a graded series of alcohols, embedded in paraffin blocks and cut at a thickness of 5 µm. Sections were dewaxed in xylene and rehydrated through a series of graded alcohols. Then, sections were incubated in the solution consisted of two volumes of 50% silver nitrate (Tianjin Tiangan Chemical Technology Development Co., Ltd., Tianjin, China) and one volume of 1% formic acid (Shanghai Macklin Biochemical Co., Ltd., Shanghai, China) in 2% gelatin (Maokangbio, Shanghai, China) for 1 h in the dark at room temperature. Tibial samples were then washed in PBS for 5 min and incubated in 5% aqueous sodium thiosulfate solution for 10 min. Sections were rinsed in PBS, dehydrated in alcohols,

cleared in xylene, and mounted for observation. The canalicular density and length of osteocytes in tibial bone matrix were determined using ImageJ.

Statistical Analysis

Statistical analyses of data were performed using the SPSS software (version 22, SPSS Inc.). All data are presented as the mean \pm S.D ($n=8$ in each group). Unpaired, two-tailed student's t test was used to compare the difference between the contralateral and loaded tibiae. Differences were considered statistically significant at $***P<0.001$.

Results

The results of Ploton silver staining (as shown in the figure below) demonstrated that uniaxial cyclic compressive loading on the tibia for 2 weeks with the peak strain of $1455 \mu\epsilon$ resulted in a significant increase in canalicular length and density in normal mice as compared with the contralateral control tibia ($P<0.001$, +36.9% in canalicular length and +35.3% in canalicular density).

Figure 1. Exogenous cyclic loading induced increases in osteocyte canalicular length and density in mice. The left tibia was subjected to daily physiological

cyclic compressive loading with 1200 cycles/day (with the waveform of cyclic ramp loading at 4 Hz) for 2 weeks, and the contralateral right tibia was not subjected to cyclic loading and served as a control. Then, decalcified and embedded tibia specimens were stained with Ploton silver for visualizing the morphology of the osteocyte canalicular network. Graphs represent mean \pm SD ($n=8$ mice per group). *** $P<0.001$ by Student's t test. Scale bars: 20 μm .

REVIEWERS' COMMENTS

Reviewer #2 (Remarks to the Author):

I am happy with the response by the authors and the additions to the manuscript.

The replications of the findings regarding the increase in canalicular length within a 2 week time frame support the previously reported results in the manuscript. Therefore I deem the paper suitable for publication.